# A twist defect mechanism for ATP-dependent translocation of nucleosomal DNA

Jessica Winger, Ilana M Nodelman, Robert F Levendosky, Gregory D Bowman*

T.C. Jenkins Department of Biophysics, Johns Hopkins University, Baltimore, United States

**Abstract** As superfamily 2 (SF2)-type translocases, chromatin remodelers are expected to use an inchworm-type mechanism to walk along DNA. Yet how they move DNA around the histone core has not been clear. Here we show that a remodeler ATPase motor can shift large segments of DNA by changing the twist and length of nucleosomal DNA at superhelix location 2 (SHL2). Using canonical and variant 601 nucleosomes, we find that the *Saccharomyces cerevisiae* Chd1 remodeler decreased DNA twist at SHL2 in nucleotide-free and ADP-bound states, and increased twist with transition state analogs. These differences in DNA twist allow the open state of the ATPase to pull in ~1 base pair (bp) by stabilizing a small DNA bulge, and closure of the ATPase to shift the DNA bulge toward the dyad. We propose that such formation and elimination of twist defects underlie the mechanism of nucleosome sliding by CHD-, ISWI-, and SWI/SNF-type remodelers.
DOI: https://doi.org/10.7554/eLife.34100.001

## Introduction

Chromatin remodelers are members of the extensive superfamily 2 (SF2) group of ATPase motors found in all kingdoms of life (*Flaus and Owen-Hughes, 2001*; *Flaus et al., 2006*). SF2 ATPases possess a bi-lobed architecture, with a central ATP-binding pocket defined by two RecA-like domains (reviewed in *Singleton et al., 2007*). The occupancy of the central pocket – whether empty, ADP-bound or ATP-bound – determines the preferred conformation for the two lobes of the motor. Like the architecturally similar SF1 ATPases, both lobes of the motor coordinate in binding to DNA or RNA, with conformational changes driven by cycles of ATP binding and hydrolysis enabling many SF2 enzymes to translocate in an inchworm fashion along nucleic acids (*Singleton et al., 2007*).

Chromatin remodelers are specialized for altering the organization of DNA on the nucleosome. Canonical nucleosomes consist of two copies each of the four core histones (H2A, H2B, H3, and H4) wrapped by ~146 base pairs (bp) of duplex DNA (reviewed in *McGinty and Tan, 2015*). Several chromatin remodelers have been shown to ratchet DNA past the histone core in single bp steps, known as nucleosome sliding, which is consistent with the basic inchworm mechanism of DNA translocation (*Deindl et al., 2013*; *Harada et al., 2016*).

One of the earliest models for nucleosome motion suggested that DNA could be shifted past the histone core by twist diffusion, where transfer of one or a small number of bps from one DNA segment to the next results in DNA displacement (*van Holde and Yager, 2003*; *van Holde and Yager, 1985*). Each segment of nucleosomal DNA that gains or loses a bp relative to the canonical nucleosome experiences a change in both DNA length and twist and is referred to as a twist defect. In addition to explaining heat-induced movement of the histone core along DNA, it was proposed that diffusion of twist defects could explain nucleosome movement by chromatin remodelers (*van Holde and Yager, 2003*). Given the helical nature of the double helix, DNA translocation by a chromatin remodeler would necessarily impart twist into segments of nucleosomal DNA adjacent to the motor,

*For correspondence:
gdbowman@jhu.edu

Competing interests: The authors declare that no competing interests exist.

**eLife digest** DNA is shaped like a spiral staircase, twisting around itself to create a double helix. This results in a long string-like molecule that needs to be carefully packaged to fit inside the cells of organisms as diverse as fungi or humans. This packaging process starts when a portion of DNA tightly wraps around a spool-like core of proteins called histones. The resulting structure is known as a nucleosome. Like the beads on a necklace, nucleosomes exist at regular intervals along DNA.

The DNA sequence around the histones cannot be accessed by a cell, and so the nucleosomes need to be 'shifted' along DNA to free up the genetic information. Enzymes known as chromatin remodelers perform this role by binding to a nucleosome, and then using energy to fuel a change in their structure that makes them 'crawl' on DNA like an inchworm. During this process, chromatin remodelers slide nucleosomes along the DNA, but it was unclear how exactly the inchworm motions pushed DNA around the histones.

Here, Winger et al. look into the details of this mechanism by focusing on the chromatin remodeler Chd1, which is conserved from yeast to humans. Experiments show that, first, the enzyme slightly untwists the DNA double helix; this untwisting causes the DNA to pucker a little on the nucleosome. The puckering creates tension and 'pulls' DNA towards the remodeler. Then, Chd1 changes its structure and twists DNA in the opposite direction, which forces the puckered DNA onto the other side of the remodeler. This extra bit of DNA then propagates around the rest of the nucleosome, like the wave created by flicking the end of a long rope. This sheds light on how these enzymes can ratchet DNA past the histones.

As the gatekeepers of our genetic information, chromatin remodelers are key to the health of the cell – in fact, they are often affected in cancers. The work by Winger et al. creates a framework that will help to understand how exactly chromatin remodelers help cells access the genetic information that the body needs to function properly.

DOI: https://doi.org/10.7554/eLife.34100.002

stimulating propagation of single bps onto and off of the nucleosome. Indeed, experimental support for twist diffusion comes from the observations that chromatin remodelers shift DNA off the nucleosome in single bp steps (*Deindl et al., 2013*; *Harada et al., 2016*) and can induce and are sensitive to torsional strain (*Gavin et al., 2001*; *Havas et al., 2000*).

In contrast to INO80, which appears to translocate on DNA near the edge of the nucleosome around superhelix location 6 (SHL6) (*Ayala et al., 2018*; *Brahma et al., 2017*; *Eustermann et al., 2018*), Chd1, ISWI, and SWI/SNF-type remodelers translocate on nucleosomal DNA at the internal SHL2 site, located ~20 bp from the nucleosome dyad (*Deindl et al., 2013*; *McKnight et al., 2011*; *Saha et al., 2005*; *Schwanbeck et al., 2004*; *Zofall et al., 2006*). Recent work has revealed the organization of Chd1 domains on the nucleosome in the ADP·BeF$_3^-$ state, which has defined the positioning and orientation of ATPase motor domains on nucleosomal DNA at SHL2 (*Farnung et al., 2017*; *Nodelman et al., 2017*; *Sundaramoorthy et al., 2018*). However, the underlying mechanism of nucleosome sliding – how the remodeler ATPase stimulates DNA translocation from this internal site on the nucleosome – has remained elusive.

Here we dissect conformational changes coupled to the ATP binding and hydrolysis cycle of the Chd1 remodeler, which reveals an unexpected process of remodeler-catalyzed DNA translocation. Using site-specific cross-linking, we show how both Chd1 domains and nucleosomal DNA change position in response to different nucleotide-bound states of the ATPase motor. Remarkably, Chd1 can shift the outer turn or gyre of nucleosomal DNA by 1–3 bp in nucleotide-free (apo) and ADP-bound states, consistent with stabilization of a small DNA bulge at SHL2. The shift of entry DNA upon Chd1 binding occurs in a sequence-selective fashion, and correlates with the phasing strength for nucleosomal DNA surrounding the SHL2 binding site of the ATPase motor. Using a modified Widom 601 nucleosome positioning sequence, we demonstrate that Chd1 binding at SHL2 alters the twist of nucleosomal DNA in a nucleotide-dependent fashion. These findings suggest that nucleosome sliding by Chd1 occurs by creating and expelling twist defects at SHL2, providing insight into the mechanism of chromatin remodeling.

## Results

### Interactions between the ATPase motor of Chd1 and nucleosomal DNA are sensitive to nucleotide state and DNA sequence

In previous work, we identified the arrangement of Chd1 domains on the nucleosome using site-specific cross-linking with 4-azidophenacyl bromide (APB) (*Nodelman et al., 2017*). Cross-linking requires van der Waals contact between the reactive nitrene moiety of APB and its DNA target, which corresponds to a reach of ~11 Å from the Cα of the APB-modified cysteine (*Pendergrast et al., 1992*). Since cross-links do not necessarily represent the closest DNA contacts, multiple independent cross-links are required to determine the position and orientation of a protein domain with respect to DNA. Our modeling, which was based on nine different cross-linking sites, yielded a structurally similar placement of Chd1 domains as cryoEM complexes solved in an ADP·BeF$_3^-$ state (*Farnung et al., 2017*; *Sundaramoorthy et al., 2018*). While ADP·BeF$_3^-$ has been cited as a ground-state mimic of ATP, the BeF$_3^-$ moiety as the gamma phosphate mimic can occupy a range of positions relative to ADP, depending on the system and conditions, and thus can also resemble a hydrolysis intermediate (*Kagawa et al., 2004*; *Thomsen and Berger, 2009*). As an SF2-type translocase, the ATPase motor of Chd1 is expected to oscillate between open and closed states during nucleosome sliding. In this work, we use five cross-linking sites on the ATPase motor to monitor how contacts with DNA change in different nucleotide states. Single cysteine substitutions were located in lobe 1 (N459C and E493C) and lobe 2 (V721C, N650C, and S770C; *Figure 1A*). A summary of where these sites cross-link in the context of the nucleosome is shown in *Figure 1—figure supplement 1*.

Chd1 interacts with both strands of the minor groove at SHL2 (*Farnung et al., 2017*; *Nodelman et al., 2017*). Based on comparisons to SF1 and SF2 ATPases that translocate along single stranded DNA or RNA, these two strands are referred to as the 'tracking' and 'guide' strands. Like the SWI/SNF-type remodeler RSC, Chd1 is positioned to translocate along the tracking strand in a 3' to 5' direction (*Farnung et al., 2017*; *Nodelman et al., 2017*; *Saha et al., 2005*). As the SHL2 site is internally located, it is expected that the ATPase motor must remain in place on the nucleosome as it shifts DNA past the histone core. Consistent with this notion, cross-links for both ATPase lobes indicate a relatively fixed position of the motor at SHL2 in apo, ADP, and ATP-mimic states (*Figure 1* and *Figure 1—figure supplement 1*).

Cross-linking was expected to vary in a nucleotide-dependent fashion, since the two lobes of SF2 ATPases are tightly packed together in ATP-bound states and are more open in apo and ADP states. However, we also observed cross-linking differences between the two sides of the nucleosome. The Widom 601 nucleosome positioning sequence is non-palindromic, and one sequence feature of the 601 that is asymmetrically distributed on the two sides are periodic TpA dinucleotides (TA steps) (*Chua et al., 2012*; *Lowary and Widom, 1998*). Phased relative to other sequence elements of the 601, the TA steps allow DNA to bend more easily around the histone octamer, with unique geometry that allows for tighter histone binding to both the H3-H4 tetramer and H2A-H2B dimers (*Chua et al., 2012*; *Ngo et al., 2015*). Given the asymmetric distribution of TA steps, we refer to the two sides of the 601 as TA-poor and TA-rich, with the TA-poor side denoted with positive (+) numbering and the TA-rich side with negative (-) numbering (see *Figure 1—figure supplement 2*).

The two lobes of the ATPase motor displayed different sensitivities with changes in nucleotide state. Variation in nucleotide conditions produced relatively little change in lobe 1 cross-linking. Both N459C and E493C cross-linked to the same sites in all nucleosome conditions, though stronger cross-links were often observed in ADP·BeF$_3^-$ conditions, possibly reflecting higher affinity binding (*Figure 1B,C* and *Figure 1—figure supplement 3A*). E493C cross-linked to both strands on both sides of the nucleosome, with three of the four sites relatively insensitive to nucleotide. The fourth site, a doublet on the tracking strand of the TA-poor side, did show subtle but distinct variation in band intensity in AMP-PNP compared with apo/ADP conditions.

In contrast to lobe 1, all cross-links from lobe 2 were sensitive both to the nucleotide state and DNA sequence. V721C (lobe 2) failed to cross-link in apo/ADP conditions on either side of the nucleosome, and cross-linking on the TA-poor side was also severely reduced in AMP-PNP (*Figure 1D*). However, the other two lobe 2 cross-linking positions, N650C and S770C, did cross-link in apo and ADP conditions, indicating that lobe 2 was indeed engaged with nucleosomal DNA in all four nucleotide conditions. Like V721C, the other two lobe 2 cross-linking sites were also highly sensitive to

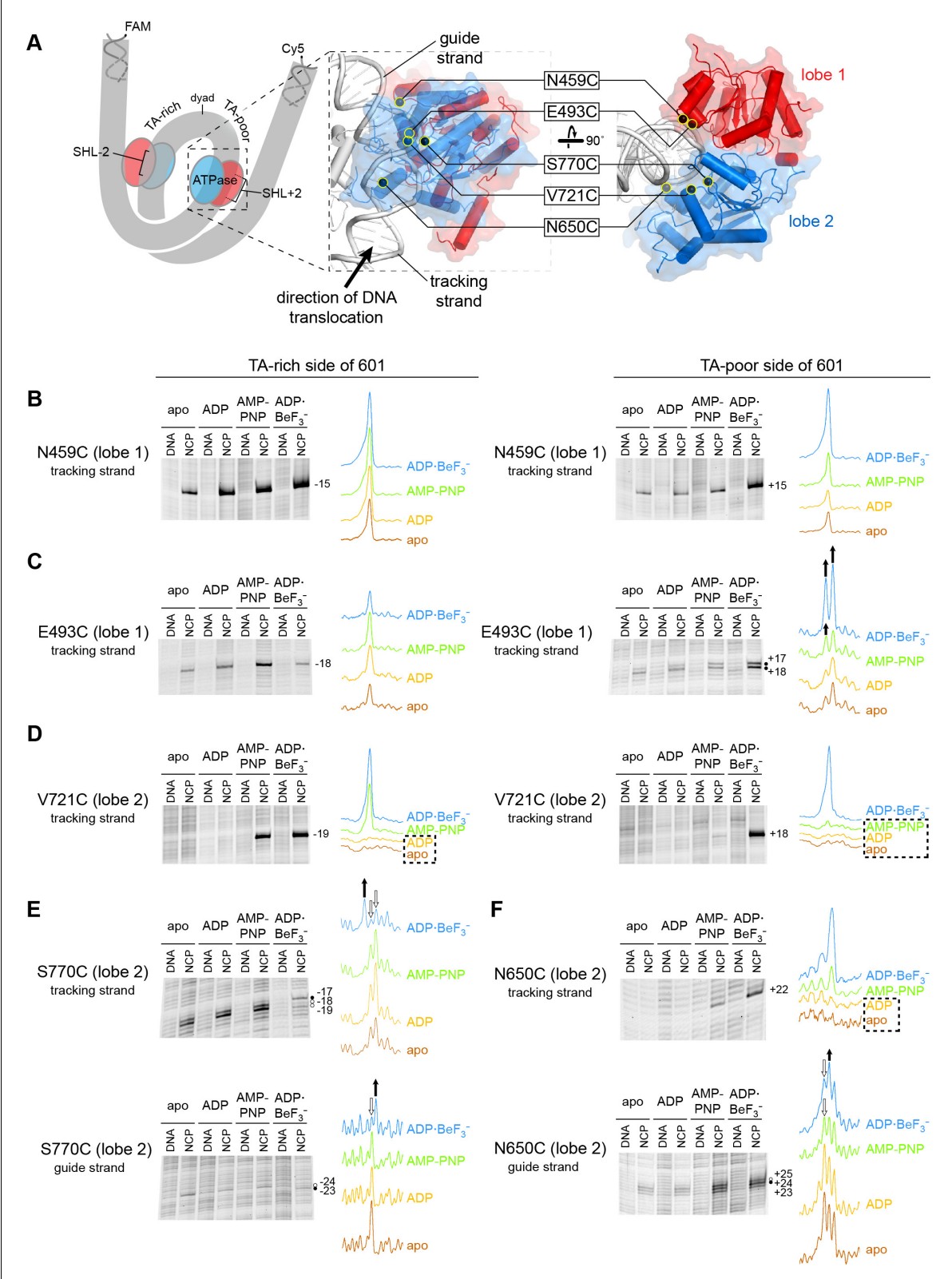

**Figure 1.** Interactions between the ATPase motor of Chd1 and nucleosomal DNA are sensitive to both nucleotide state and DNA sequence. (**A**) Overview of the Chd1 ATPase motor on the nucleosome. A cartoon schematic (left) shows the superhelical wrap of nucleosomal DNA, with an ATPase motor (blue/red) at each of the superhelix location (SHL) 2 sites on either side of the nucleosome. The expanded views on the right show the molecular surface of the ATPase motor in complex with nucleosomal DNA, based on a cryoEM complex (*Farnung et al., 2017*). The five positions individually

*Figure 1 continued on next page*

*Figure 1 continued*

substituted with cysteine for APB cross-linking are highlighted with yellow circles, and the tracking and guide strands of DNA are indicated. The PyMOL visualization package (RRID:SCR_000305) was used for molecular rendering. (B-F). Representative data showing cross-linking to nucleosomal DNA on either the TA-rich side (left) or TA-poor side (right) of the 601 sequence, in different nucleotide conditions. Identical cross-linking reactions were performed with 300-450 nM Chd1 variant with either 150 nM canonical 601 nucleosomes with 40 bp of flanking DNA on each side ('NCP') or 150 nM naked DNA ('DNA'), to ensure observed cross-links were nucleosome-specific. Bands that increase or decrease in cross-linking efficiency for ADP·BeF$_3^-$ compared to other conditions are indicated with black or white circles, respectively. Scans of each lane are shown as intensity plots to the right of each gel, with arrows highlighting bands that showed relative increases (black) or decreases (white) for ADP·BeF$_3^-$ conditions. Dotted boxes highlight nucleotide conditions where cross-links were severely reduced or not detected. Numbering refers to the distances (nt) of cross-linked sites from the 601 dyad (positive numbers for TA-poor side and negative numbers for TA-rich side). These data are representative of 4 or more experiments. Extended gel images are shown in *Figure 1—figure supplement 3*.

DOI: https://doi.org/10.7554/eLife.34100.003

The following figure supplements are available for figure 1:

**Figure supplement 1.** Site-specific cross-linking visualized on the Chd1-nucleosome structure.

DOI: https://doi.org/10.7554/eLife.34100.004

**Figure supplement 2.** DNA sequences of 601 variants used in this study.

DOI: https://doi.org/10.7554/eLife.34100.005

**Figure supplement 3.** Extended gel images of Chd1 cross-linking reactions.

DOI: https://doi.org/10.7554/eLife.34100.006

**Figure supplement 4.** Chd1(N650C) cross-linking on 601[swap SHL2.5/3.5] nucleosomes yielded a distinct nucleotide-dependent pattern.

DOI: https://doi.org/10.7554/eLife.34100.007

DNA sequence: N650C failed to cross-link to the tracking strand on the TA-rich side, whereas S770C failed to cross-link to the tracking strand on the TA-poor side (*Figure 1—figure supplement 3B*). The cross-linking patterns for both N650C and S770C also varied depending on the bound nucleotide (*Figure 1E,F*). This sensitivity of lobe 2 cross-linking to DNA sequence suggests that the conformation and/or stability of DNA strongly influence the engagement of lobe 2 at SHL2.

In two instances, cross-linking to the guide strand seemed to be somewhat at odds with expected structural interpretations. S770C made weak but significant cross-links to the guide strand 23 and 24 nt from the dyad (*Figure 1E* and *Figure 1—figure supplement 3B*). In contrast with other cross-links mapped onto the cryoEM structure, these guide strand cross-links span a distance of ~17–21 Å and are therefore significantly longer than expected for APB (*Figure 1—figure supplement 1*). This discrepancy likely reflects conformational variability in the protein and/or DNA not captured by the structure. A second instance was guide strand cross-linking to N650C. In agreement with lobe 2 moving toward the dyad upon cleft closure, both N650C and S770C showed stronger cross-linking closer to the dyad in ADP·BeF$_3^-$ compared with apo, ADP and AMP-PNP conditions (*Figure 1E,F*). However, a different nucleotide-dependent cross-linking pattern was obtained for N650C with nucleosomes possessing a different DNA sequence. Using a variant of the Widom 601 sequence where segment 24–39 on each side of the dyad had been swapped (called 601[swap SHL2.5/3.5], see *Figure 1—figure supplement 2*), stronger cross-linking was instead observed closer to the dyad in apo/ADP conditions (*Figure 1—figure supplement 4*). While these findings indicate that physical interpretations of each cross-linking reaction should be taken with caution, the combined results suggest that lobe 2 and/or the guide strand are dynamic and sensitive to sequence and structural perturbations.

## Chd1 binding to 601 nucleosomes in apo/ADP states can shift entry DNA by ~2 bp

We and others have used APB cross-linking of single cysteine-substituted histones to monitor changes in nucleosome positioning during ATP-dependent sliding by chromatin remodelers. In the course of our studies, we noticed a two nt shift in cross-linking upon addition of Chd1 under conditions where ATP was absent (*Figure 2—figure supplement 1*). This shift was observed with H2B (S53C), which cross-links to DNA 53 nt on either side of the nucleosome dyad of the Widom 601 (*Kassabov et al., 2002*) (*Figure 2A*). To investigate this phenomenon further, we monitored H2B (S53C) cross-linking in the presence and absence of Chd1 in four nucleotide conditions (apo, ADP, AMP-PNP, and ADP·BeF$_3^-$). No changes in H2B(S53C) cross-linking were observed with AMP-PNP

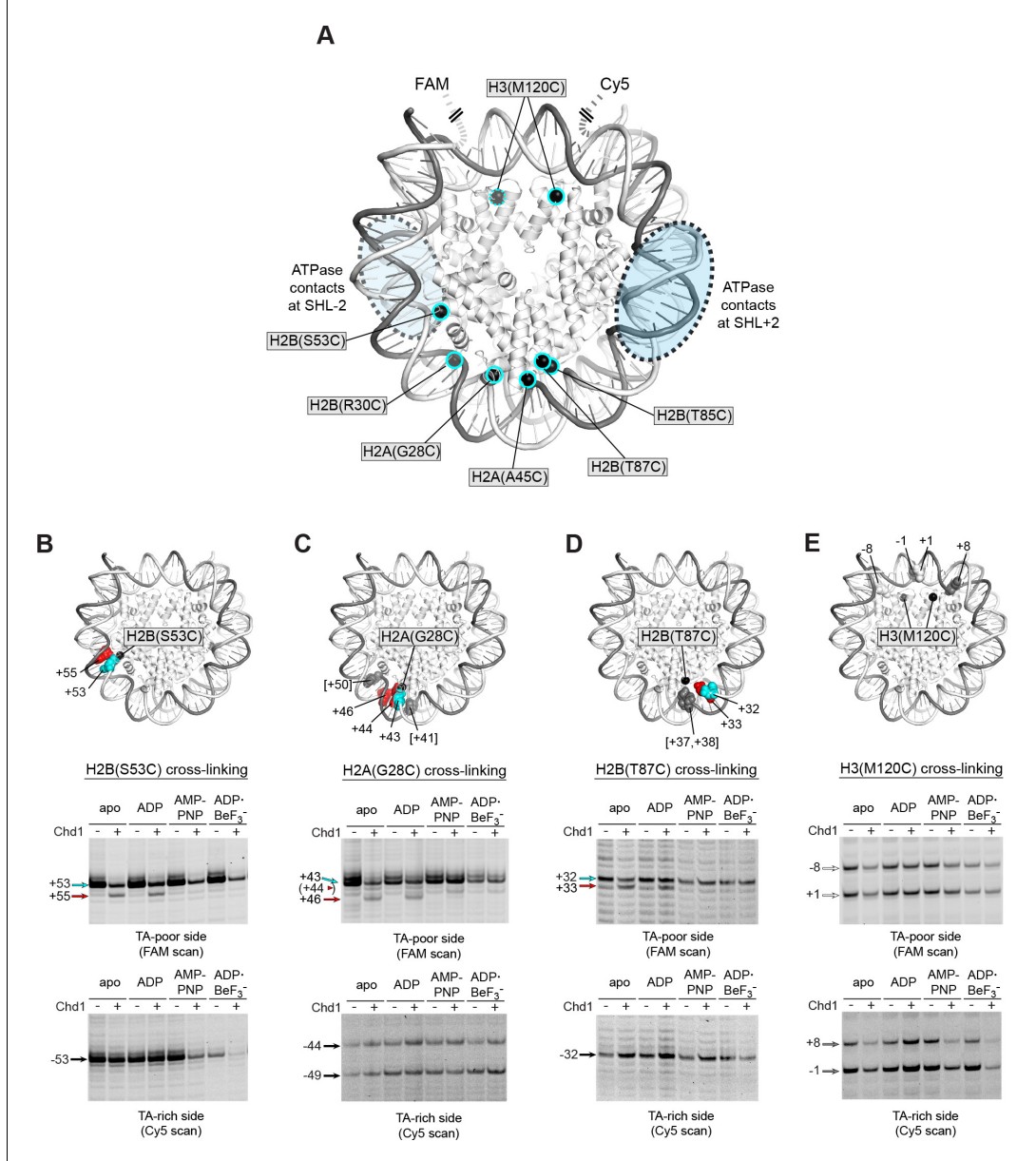

**Figure 2.** Chd1 binding in apo and ADP conditions shifts nucleosomal DNA on the TA-poor side of the 601 sequence. (A) Molecular representation of a 601 nucleosome (*Makde et al., 2010*), indicating sites of cysteine substitution for site-specific cross-linking. Blue dotted ovals indicate the regions bound by the Chd1 ATPase motor at each SHL2 site (*Farnung et al., 2017*; *Nodelman et al., 2017*; *Sundaramoorthy et al., 2018*). (B) Histone H2B (S53C) cross-linking reactions in the presence and absence of Chd1. PyMOL representation shows cross-links on the TA-poor side of the nucleosome that occur for the nucleosome alone (cyan) and in the presence of Chd1 (red). Cross-linking was performed using 150 nM canonical 601 nucleosomes with 40 bp flanking DNA on each side (40N40), in the presence or absence of 600 nM Chd1. Cross-linked products were separated on urea denaturing gels and visualized by FAM or Cy5 fluorescence. Numbering refers to the distances of cross-linked sites from the 601 dyad (bp). (C) Histone H2A(G28C) cross-linking reactions, performed as described in B. In the PyMOL representation, sites of cross-linking that occur on the complementary strand are indicated by gray spheres with numbering in square brackets (see *Figure 2—figure supplement 2B* for +41 and +50 cross-linking sites). (D) Histone H2B(T87C) cross-linking reactions, performed as described in B. As for (C), cross-links that occur on the complementary strand [+37/+38] are shown as gray spheres in the PyMOL representation (*Figure 2—figure supplement 2C*). (E) Histone H3(M120C) cross-linking reactions, performed as described in B. All cross-linking experiments are representative of 3 or more experiments. Extended gel images are shown in *Figure 2—figure supplement 2D*.
DOI: https://doi.org/10.7554/eLife.34100.008

The following figure supplements are available for figure 2:

**Figure supplement 1.** Titration of Chd1 for H2B(S53C) cross-linking reaction.
DOI: https://doi.org/10.7554/eLife.34100.009

*Figure 2 continued on next page*

*Figure 2 continued*

**Figure supplement 2.** Extended gel images for histone mapping of Widom 601 nucleosomes.
DOI: https://doi.org/10.7554/eLife.34100.010
**Figure supplement 3.** Treatment with hexokinase confirms that ADP is sufficient for supporting a Chd1-dependent shift in H2B(S53C) cross-linking.
DOI: https://doi.org/10.7554/eLife.34100.011
**Figure supplement 4.** Histone mapping with other single cysteine histone variants.
DOI: https://doi.org/10.7554/eLife.34100.012

and ADP·BeF$_3^-$, whereas a two nt shift was apparent in a Chd1-dependent manner for both apo and ADP conditions (*Figure 2B* and *Figure 2—figure supplements 2* and *3*). This shift was only apparent on the TA-poor side of the Widom 601 sequence.

We interpreted this new cross-link as a Chd1-induced change in histone-DNA contacts. To determine whether other regions of nucleosomal DNA were similarly affected by Chd1 binding, we generated and tested the cross-linking characteristics of six other single-cysteine histone variants (*Figure 2A*). Although H2B(R30C) failed to cross-link, the other five variants yielded specific cross-linking patterns (*Figure 2—figure supplements 2* and *4*). Of these, cross-linking for both H2A (G28C) and H2B(T87C) was clearly affected by Chd1, showing patterns similar to H2B(S53C) at roughly 10 nt intervals from each other. On the TA-poor side of the 601 nucleosome, H2A(G28C) cross-linked primarily +43 nt from the dyad, and addition of Chd1 in apo and ADP conditions yielded novel cross-linking at +46 and more weakly at +44 (*Figure 2C*). For H2B(T87C), which cross-linked +32 nt from the dyad, addition of Chd1 produced a new cross-link at +33 (*Figure 2D*). The shifts in cross-linking from these three sites were all nucleotide specific, occurring only in apo and ADP conditions, and also were only apparent on the TA-poor side of the nucleosome.

Despite their proximity to H2B(S53C), H2A(G28C) and H2B(T87C), no clear Chd1-dependent changes in cross-linking were observed for H2A(A45C) or H2B(T85C). For H2B(T85C), the lack of a Chd1-dependent change may have been due to sensitivity to the 601 sequence, since cross-linking was much more pronounced on the TA-rich compared to the TA-poor side (*Figure 2—figure supplement 4*). Consistent with this, the neighboring H2B(T87C) site was strongly affected by the 601 sequence, showing either single or double cross-links to each strand depending on the side of the nucleosome (*Figure 2—figure supplement 2C*). H2A(A45C), however, produced strong symmetric cross-links on each side of the nucleosome (*Figure 2—figure supplement 4*). Curiously, all of the Chd1-dependent cross-links occurred on one strand, and H2A(A45C) cross-linked to the complementary DNA strand at +40. Like H2A(A45C), H2B(T87C) also cross-linked to the complementary strand, at +37/+38. This doublet did subtly change in intensity with Chd1 in apo and ADP conditions, yet no new cross-links were observed (*Figure 2—figure supplement 2C*). This correlation raises the intriguing possibility that the histone-DNA interactions of the two strands may be differently affected by Chd1 binding. Alternatively, differences in cross-linking may have arisen from distortion of the histone core. The hydrophobic cores of H2A and H4 have been shown to be affected by binding to the Snf2h remodeler (*Sinha et al., 2017*), and recent cryoEM analyses have revealed a remarkable flexibility of the histone core (*Bilokapic et al., 2018a*; *Bilokapic et al., 2018b*). A distortion of the histone dimer might therefore be responsible for the non-uniform cross-linking response to Chd1 binding.

In addition to the cross-linking histone variants described above, we also tested H3(M120C), which yields double cross-links at the dyad (*Hota et al., 2013*). H3(M120C) cross-linking was also unaffected by the presence of Chd1 (*Figure 2E*). As we describe below, the H3(M120C) cross-linking can show Chd1-dependent changes in certain nucleosome contexts. Therefore, here we interpret the absence of cross-link changes as a lack of Chd1-dependent changes in histone-DNA contacts around the nucleosome dyad of Widom 601 nucleosomes.

## DNA is shifted toward the same gyre SHL2 site

The nucleosome can be bound by two Chd1 molecules, one at each SHL2 site (*Nodelman et al., 2017*), and we hypothesized that the DNA shifts we observed were stimulated by Chd1 binding at one of those sites. The SHL+2 binding site is adjacent to the segment of DNA where histone-DNA cross-links changed in response to Chd1 binding. The SHL−2 site is on the opposite gyre, and therefore almost one superhelical turn of DNA away. Recent cryoEM structures, however, show that Chd1

binding at SHL2 unwraps ~20 bp of DNA on the opposite gyre (*Farnung et al., 2017*; *Sundaramoorthy et al., 2017*; *Sundaramoorthy et al., 2018*). Therefore, it is possible that the Chd1-dependent changes in cross-linking could arise from Chd1 binding at either SHL2 site. To determine which side might be responsible, we employed a previously devised biotin/streptavidin blocking strategy (*Nodelman et al., 2017*), where a biotin moiety was attached to nucleosomal DNA, 19 nt from the dyad on one side. Addition of streptavidin should occlude the biotinylated SHL2, and thus selectively allow association of Chd1 to the other SHL2 site.

Chd1 binding was monitored using the Chd1(N459C) cross-linking variant, while shifts in histone-DNA cross-linking were followed with the H2B(S53C) variant. For the nucleosome construct with biotin on the TA-rich SHL–2 site, the addition of streptavidin significantly diminished Chd1 binding to SHL–2, as expected, yet a robust Chd1-dependent shift in the H2B(S53C) cross-link was still observed (*Figure 3A*). This result indicates that the shift in histone-DNA cross-linking (+53 to +55) was accomplished by Chd1 binding SHL+2 on the same gyre. In support of this conclusion, the H2B (S53C) cross-linking shift was completely blocked on the nucleosome variant with biotin at SHL+2, on the TA-poor side (*Figure 3B*).

As observed in cryoEM structures, Chd1 unwraps ~2 turns of DNA from the nucleosome edge of the opposite gyre (*Farnung et al., 2017*; *Sundaramoorthy et al., 2017*; *Sundaramoorthy et al., 2018*). Unwrapping by Chd1 appeared to diminish the signal from the H2B(S53C) cross-linking site, because the biotin/streptavidin block consistently yielded stronger H2B(S53C) cross-linking on the side opposite where Chd1 could no longer bind (compare *Figure 3* to *Figure 2—figure supplement 1*). With a biotin block on the TA-rich side, H2B(S53C) cross-linking on the TA-poor side was stronger yet still showed the Chd1-induced shift, indicating that the change in cross-linking resulted from Chd1 binding to DNA on the same gyre.

Relative to the remodeler bound at the SHL+2 site, the Chd1-dependent cross-links all occur on the tracking strand. We propose that these shifts in cross-linking result from Chd1-dependent movement of DNA relative to the H2A/H2B dimer. From these experiments, we cannot distinguish between movement of the histone dimer and movement of DNA relative to the rest of the nucleosome. However, all of the cross-linking shifts occur in the same direction, and would correspond to movement of DNA toward the Chd1-bound SHL+2 site. We propose that Chd1 could accomplish such a shift if binding stabilized a small ~1 bp bulge of nucleosomal DNA at SHL2. Given the helical nature of duplex DNA, a small bulge would reduce local twisting of the double helix and pull DNA toward the remodeler in a corkscrew-like motion (*Figure 3C*). A corkscrew-like motion is also consistent with the ability to still observe histone-DNA cross-linking, which likely requires a canonical path of nucleosomal DNA with respect to the histone core. Experiments described in the last section further support this idea that trapping or excluding a bulge at SHL2 allows Chd1 to rotate and translocate DNA on the nucleosome.

The cryoEM structures of Chd1 and the SWI/SNF ATPase show intimate remodeler-DNA contacts at SHL2 (*Farnung et al., 2017*; *Liu et al., 2017*). In both structures, the DNA backbone is directly contacted by a tryptophan residue from the ATPase motor, and in the apo SWI/SNF structure, this contact occurs at a location of significant DNA distortion on the guide strand. This tryptophan is highly conserved across all remodeler families and therefore appears to be structurally important. To investigate the significance of this residue, we generated a Chd1(W793A) variant to see if removing this DNA contact altered the Chd1-dependent shifts in H2A/H2B cross-links. Using the double variant Chd1(W793A/N459C), we found that the W793A substitution did not interfere with ATPase binding to either the TA-rich or TA-poor SHL2 in apo conditions (*Figure 3—figure supplement 3A*). Interestingly, despite the intimate contacts between this tryptophan and DNA in the cryoEM structures, the Chd1(W793A/N459C) also produced a similar shift of the H2B(S53C) cross-link (*Figure 3—figure supplement 3B*). Unlike the loss of remodeling activity observed for the corresponding mutation of yeast SWI/SNF ATPase (W1185A) (*Liu et al., 2017*), we found only a ~2 fold slower rate of sliding compared to the wild-type background (*Figure 3—figure supplement 3C,D*). We therefore conclude that despite the conservation and central placement of W793, it is not essential for nucleosome sliding by Chd1, nor required for the Chd1-dependent DNA shift described here.

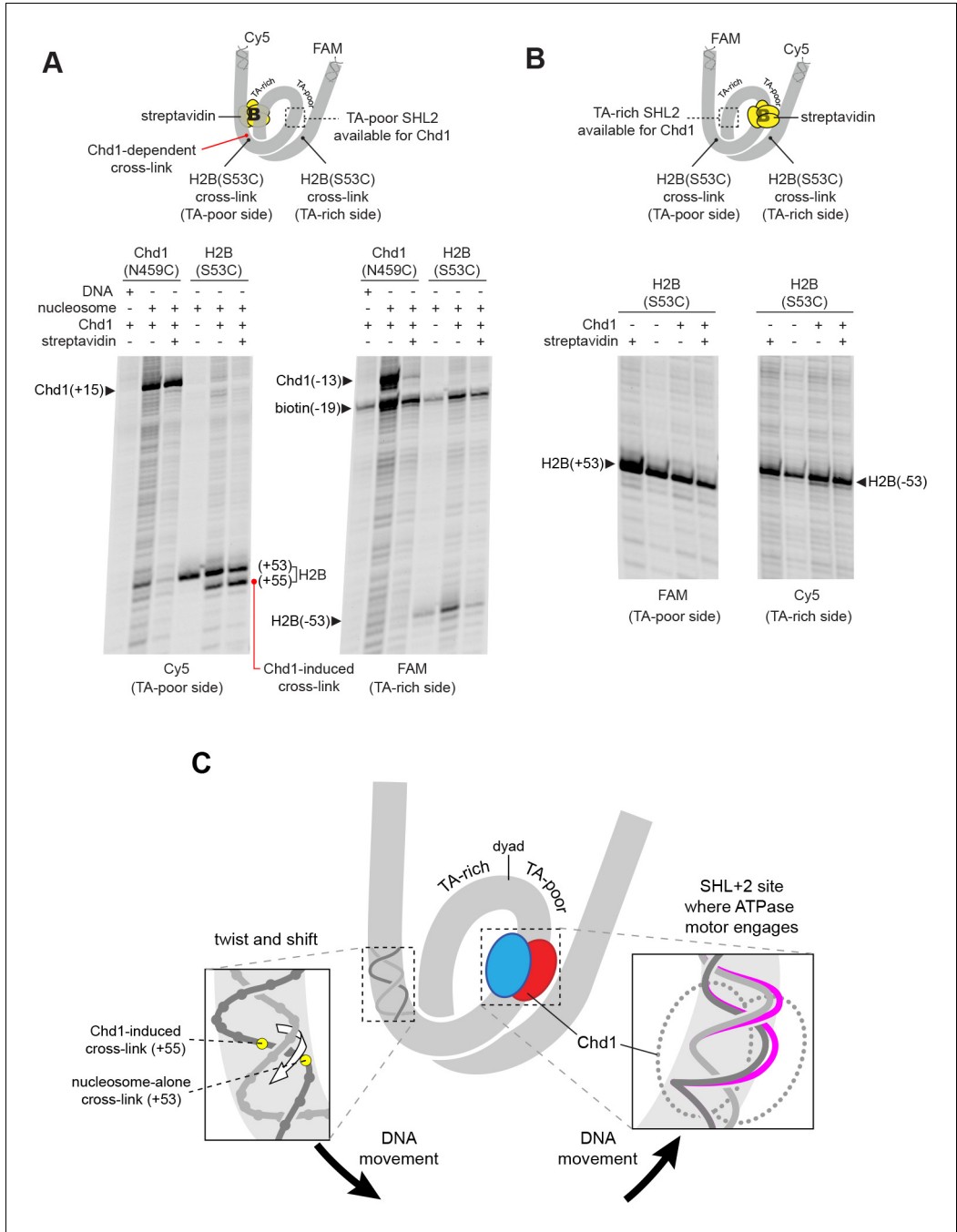

**Figure 3.** The Chd1-dependent shift in histone-DNA cross-linking correlates with Chd1 binding to the adjacent SHL2, on the same DNA gyre of the nucleosome. (**A**) The presence of a biotin/streptavidin block at the TA-rich SHL–2 site does not diminish the Chd1-dependent shift in H2B(S53C) cross-links on the TA-poor side. The nucleosome substrate for these experiments was 19-[+biotin]601-29, with a biotin moiety 19 nt from the dyad on the TA-rich SHL–2 site. Note that for this nucleosome, the FAM and Cy5 labels are on opposite sides compared to all other 601 nucleosomes used in this study. Chd1(N459C) cross-linking experiments monitored Chd1 binding at each SHL2, and H2B(S53C) experiments followed the Chd1-dependent shift in histone-DNA cross-links on the TA-poor side. The apparent shift in Chd1(N459C) cross-linking site on the biotinylated strand (labeled as −13) was due in part to slower migration caused by the biotin moiety on the DNA fragment, and may not represent a true change in cross-linking position. The gels are representative of three or more experiments, and extended gel images are shown in *Figure 3—figure supplement 1A*. (**B**) The presence of a biotin moiety at the TA-poor SHL+2 site prevents the Chd1-dependent shift in H2B(S53C) cross-linking on the TA-poor side. The nucleosome substrate for these experiments was 40-601[+biotin]-19, with a biotin moiety 19 nt from the dyad on the TA-poor SHL+2 site.

*Figure 3 continued on next page*

*Figure 3 continued*

The absence of the H2B(S53C) shift likely stems from poor binding due to the biotin, as shown in **Figure 3—figure supplement 2**. The gels are representative of three or more experiments, and extended gel images are shown in **Figure 3—figure supplement 1B**. (C) A model for how Chd1 binding at SHL+2 may be coupled to shifts in DNA cross-linking.

DOI: https://doi.org/10.7554/eLife.34100.013

The following figure supplements are available for figure 3:

**Figure supplement 1.** Extended gel images for H2B(S53C) cross-linking experiments with biotinylated 601 nucleosomes.

DOI: https://doi.org/10.7554/eLife.34100.014

**Figure supplement 2.** A biotin moiety on the TA-poor SHL2 interferes with Chd1 binding, independently of streptavidin.

DOI: https://doi.org/10.7554/eLife.34100.015

**Figure supplement 3.** Mutation of a conserved tryptophan (W793A) in the ATPase core of Chd1 only modestly slows the rate of nucleosome sliding and does not interfere with the two nt shift of H2B(S53C).

DOI: https://doi.org/10.7554/eLife.34100.016

## DNA shifts induced by Chd1 binding are sensitive to the DNA sequence at SHL3

Despite Chd1 binding to both SHL2 sites, appearance of the shifted H2B(S53C), H2A(G28C), and H2B(T87C) cross-links were notably asymmetric, observed only on the TA-poor side of the 601. We hypothesized that some sequence element(s) of the 601 were responsible for either allowing Chd1-induced shifts on the TA-poor side, or preventing shifts on the TA-rich side. To explore this idea, we tested several variants of 601, where sequence segments were flipped, replaced, or swapped (**Figure 4A** and **Figure 1—figure supplement 2**).

The binding site for the Chd1 ATPase motor at SHL2 extends between 15–25 bp from the nucleosome dyad. To determine if the segment bound by Chd1 is responsible for the observed asymmetry, we tested a 601 variant where the central 61 bp was flipped, thus exchanging the 30 bp on either side of the dyad (called 601[dyad flip$_{61}$]). Despite changing the DNA sequence where Chd1 binds at each SHL2 site, this flip did not alter the Chd1-dependent shift in H2B(S53C) cross-linking (**Figure 4B** and **Figure 4—figure supplement 1**). To see if the segment adjacent to the Chd1 binding site had an impact, we copied the segment from 24 to 39 bp on the TA-rich side to the same region on the TA-poor side (called 601[duplicate SHL–2.5/–3.5]). Interestingly, this replacement appeared to prevent the shift in H2B(S53C) cross-linking on the TA-poor side (**Figure 4B** and **Figure 4—figure supplement 1**). We note that this replacement did not alter the DNA sequence surrounding the H2B(S53C) cross-link on the TA-poor side, which is 53 nt from the dyad.

Given the interference observed from the 24–39 bp segment of the TA-rich side of the nucleosome, we analyzed a variant where the 24–39 bp segments on each side of the Widom 601 sequence simultaneously replaced each other (called 601[swap SHL2.5/3.5]). Like the duplicated variant, this swap appeared to block a shift in cross-linking on the TA-poor side, but did yield a Chd1-dependent shift on the TA-rich side by one nt (**Figure 4B** and **Figure 4—figure supplement 1**). These results suggest that the 24–39 bp region on the TA-poor side was sufficient for permitting a Chd1-dependent shift in entry DNA.

As described above, the asymmetry of the Widom 601 sequence also influenced the cross-linking patterns for the Chd1 ATPase motor to nucleosomal DNA (**Figure 1**). We suspected that some of these sequence-dependent differences also arose from the 24–39 bp region on either side of the dyad. To investigate this idea, we performed Chd1 cross-linking with the 601[swap SHL2.5/3.5] variant. Chd1(V721C) in particular was markedly sensitive to DNA sequence in AMP-PNP conditions, showing robust cross-linking on the TA-rich side of the Widom 601 yet virtually no cross-linking on the TA-poor side. Interestingly, swapping the 24–39 bp regions partially reversed this effect, showing a significant increase in cross-linking with AMP-PNP on the TA-poor side and a decrease in TA-rich side cross-linking (**Figure 4—figure supplement 2**). Importantly, Chd1(V721C) cross-links 19 nt from the dyad and is therefore not directly affected by the sequence changes from swapping the 24–39 segment. These results suggest that the stability or dynamics of the ATPase motor at SHL2 is affected by the sequence of the 24–39 bp segment of the nucleosome.

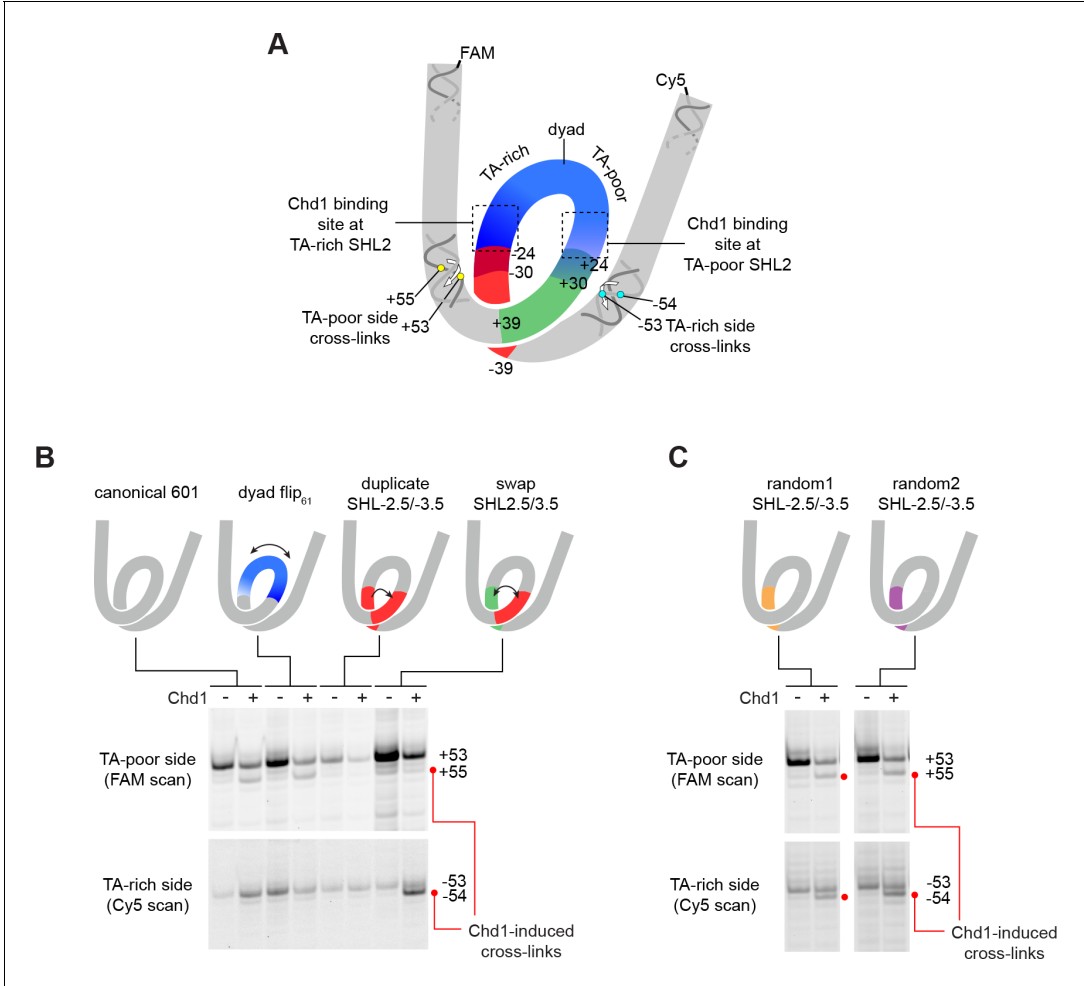

**Figure 4.** The sequence of DNA 24–39 bp from the dyad can restrict or permit shifts in H2B(S53C) cross-linking upon Chd1 binding. (**A**) Schematic representation of the nucleosome, highlighting sequence elements of the canonical Widom 601 sequence that are altered in 601 variants. (**B**) The 24–39 bp segment of the canonical 601 is responsible for allowing or blocking Chd1-dependent shifts in H2B(S53C) cross-linking. Shown are cross-linking experiments for the canonical Widom 601 and three 601 variants, carried out with 150 nM 40N40 nucleosomes and 600 nM Chd1 under apo conditions. Schematic cartoons indicate the region of 601 altered for each nucleosome variant. (**C**) Replacing the 24–39 bp segment on the TA-rich side of the 601 with random DNA sequences allows Chd1 binding (apo condition) to stimulate a shift in H2B(S53C) cross-linking on the TA-rich side. Each gel in B and C is representative of 4 or more experiments. The DNA sequences for all 601 variants are given in *Figure 1—figure supplement 2*. Similar experiments to those carried out in B and C but in other nucleotide conditions, are shown in *Figure 4—figure supplement 1*.
DOI: https://doi.org/10.7554/eLife.34100.017

The following figure supplements are available for figure 4:

**Figure supplement 1.** Effects of Chd1 binding in different nucleotide states on H2B(S53C) histone mapping of 601 variants.
DOI: https://doi.org/10.7554/eLife.34100.018

**Figure supplement 2.** Distinct TA-poor and TA-rich Chd1 cross-linking patterns are transposed on the 601[swap SHL2.5/3.5] nucleosome.
DOI: https://doi.org/10.7554/eLife.34100.019

With the 601 variants described so far, it remained unclear whether some unique sequence property of the 601 specifically blocked a Chd1-dependent DNA shift (on the TA-rich side), or was required to observe a Chd1-dependent DNA shift (on the TA-poor side). To address this issue, we created two additional 601 variants, where the 24–39 bp segment on the TA-rich side of the 601 was replaced with random DNA sequences (*Figure 1—figure supplement 2*). Notably, both of these variants enabled a Chd1-dependent DNA shift on the TA-rich side (*Figure 4C*). Thus, the sequence of the 24–39 bp segment on the TA-rich side of the canonical 601 prevented Chd1 binding from inducing DNA movement, and replacing this segment with two unrelated sequences was sufficient to allow a Chd1-dependent shift in DNA.

## The ability of Chd1 to shift nucleosomal DNA upon binding is not unique to the Widom 601 sequence

The experiments presented above show that the Chd1-dependent shifts depend on sequence context. To see if this effect was specific to the Widom 601 positioning sequence, we performed similar histone mapping experiments using nucleosomes made with the sea urchin 5S positioning sequence (*Figure 5—figure supplement 1*). The 5S sequence yielded a distribution of nucleosome positions, and through native gel purification we were able to enrich for a centered species (referred to as N0 in *Figure 5*). Notably, the enriched 5S species exhibited Chd1-dependent changes in both H2B(S53C) and H3(M120C) cross-linking patterns. For H2B(S53C) experiments, Chd1 promoted new faster-migrating cross-links for 5S nucleosomes in both apo and ADP conditions, similar to shifts seen for the Widom 601 (*Figure 5A*). Unlike the Widom 601 nucleosomes, however, Chd1 also altered the enriched 5S cross-linking patterns for H3(M120C) (*Figure 5B*). The H3(M120C) cross-linking pattern was nucleotide-dependent, most clearly showing differences in cross-linking intensity in apo and ADP conditions. The shifts in band intensity for both H3(M120C) cross-links on both DNA strands correlated with each other, suggesting a concerted change in DNA positioning at the nucleosome dyad. These results show that Chd1 binding alone is sufficient to affect histone-DNA contacts on either side of SHL2, suggesting that structural changes due to Chd1 binding in nucleotide-specific states can impact the global positioning of DNA on the histone core.

In contrast to the enriched species, several of the less populated 5S nucleosome positions failed to show shifts in cross-linking. These results reinforce our findings that this Chd1-dependent shift is sensitive to DNA sequence, and yet also suggest that such DNA shifts are a general phenomenon that can occur in various sequence contexts.

## Chd1 alters DNA length at SHL2 in a nucleotide-dependent fashion

Our model proposes that the Chd1-stimulated corkscrew motion of DNA is coupled to a local bulge at SHL2 (*Figure 3C*). We suspected that a requirement to twist DNA may explain why shifts were not observed on the TA-rich side of the Widom 601 sequence. The phased TA-steps on the TA-rich side give the 601 sequence high flexibility, and the precise positioning of these periodic TA steps where the DNA is most geometrically constrained makes it less costly to wrap around the histone octamer (*Chua et al., 2012*; *Ngo et al., 2015*). Twisting the DNA to form a bulge at SHL–2 would disrupt the phasing of TA steps on the TA-rich side, and would therefore be more energetically expensive compared to the TA-poor side, which has only one such positioned TA step. To test the hypothesis that a DNA bulge was coupled to DNA twisting, we designed a variant 601 with a single bp insertion to disrupt phasing of TA-steps on the TA-rich side (*Figure 6A*). This variant, with an A:T base pair insertion 22 bp from the dyad, is referred to as 601[insert(+1) TA-rich] (*Figure 1—figure supplement 2*). A prediction of our model was that Chd1 binding to this altered SHL–2 site in apo and ADP conditions would bring the TA-steps back into phase by effectively absorbing the extra twist from the single bp insertion.

Our initial experiments with 601[insert(+1) TA-rich] showed that Chd1 binding changed both the cross-linking pattern of H2B(S53C) on the TA-rich side, as well as H3(M120C) cross-linking around the dyad (*Figure 6—figure supplement 1*). The shifts in dyad cross-links were particularly intriguing, since the dyad is surrounded by the strongest histone-DNA contacts that appear to ultimately dictate nucleosome positioning (*Hall et al., 2009*). A complication of interpreting the dyad shifts, however, was that Chd1 can bind to both SHL2 sites, and therefore it was unclear if binding to one or a combination of both sites may have been responsible for the observed shifts. We therefore introduced a biotin/streptavidin block on the TA-poor side of the 601[insert(+1) TA-rich] variant (*Figure 6B*), allowing us to determine how the positioning of nucleosomal DNA might be affected by Chd1 binding to the altered SHL–2 site.

We generated two different biotinylated 601[insert(+1) TA-rich] nucleosomes to separately monitor H2B(S53C) and H3(M120C) cross-linking in the presence of streptavidin. Cross-linking reactions in the presence and absence of Chd1 in different nucleotide conditions are shown in *Figure 6C,D* and *Figure 6—figure supplement 2*. For H2B(S53C), the predominant cross-link for the nucleosome alone occurred at −52 instead of −53 (numbering refers to DNA positioning on the canonical 601), which is consistent with the +1 insertion at SHL–2 shifting the DNA sequence past the H2B(S53C)

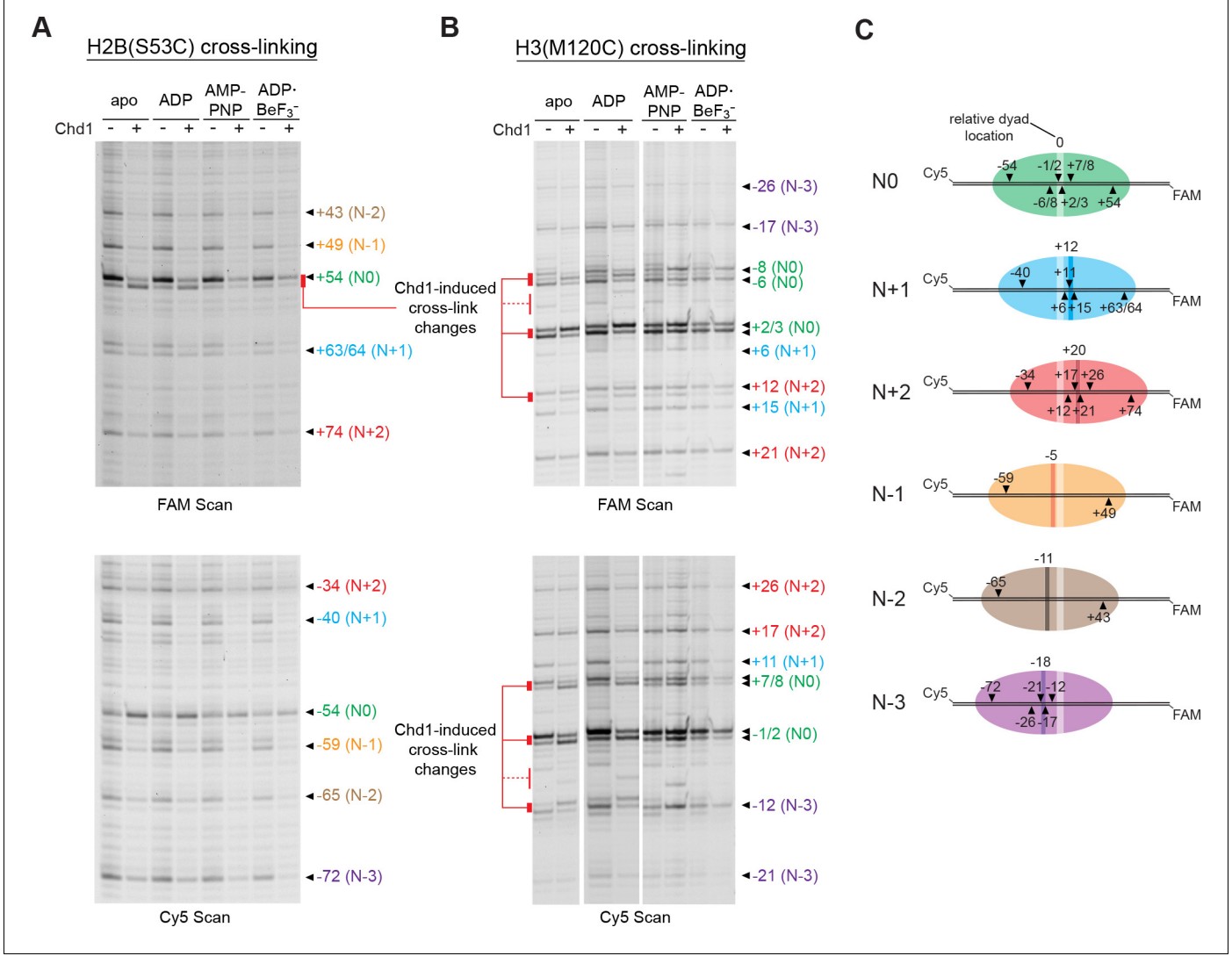

**Figure 5.** Chd1 binding alters histone-DNA contacts of 5S nucleosomes in a nucleotide-dependent manner. (A) Mapping interactions of histone H2B (S53C) with 5S nucleosomal DNA. Histone cross-linking was performed in the presence or absence of 600 nM Chd1, using a sample that was enriched in centered nucleosomes. Cross-linked products were separated on urea denaturing gels and visualized by FAM or Cy5 fluorescence. Numbering of each cross-linking site is relative to the dyad of the dominant centered species, called N0 (green). Cross-links predicted to come from the same nucleosome species are given matching colors. Prominent Chd1-induced changes in histone cross-linking are highlighted by red bars. (B) Mapping interactions of histone H3(M120C) with 5S nucleosomal DNA. Cross-linking reactions were performed similarly to those described in A. (C) Schematic representations of 5S octamer positions. Cross-links observed for H2B(S53C) and H3(M120C) are indicated with black triangles pointing to the labeled DNA strand (FAM or Cy5) where they were observed. The dyad position of the enriched N0 species is indicated by a light stripe, while the dyads for the other species are shown as dark stripes, with the approximate number of nt differing from N0 given above each stripe. The 5S sequence is given in *Figure 5—figure supplement 1*.

DOI: https://doi.org/10.7554/eLife.34100.020

The following figure supplement is available for figure 5:

**Figure supplement 1.** Sequence and dominant centered location of 5S nucleosomes.

DOI: https://doi.org/10.7554/eLife.34100.021

site by 1 bp. As shown by H3(M120C) cross-linking, the dyad for 601[insert(+1) TA-rich] nucleosomes alone was in the same location as canonical 601.

Consistent with the idea that Chd1 binding affects DNA twist on the nucleosome, addition of Chd1 influenced the length of DNA between H2B(S53C) and H3(M120C) cross-linking sites in a nucleotide-dependent manner. Interestingly, in all nucleotide conditions, Chd1 binding shifted the

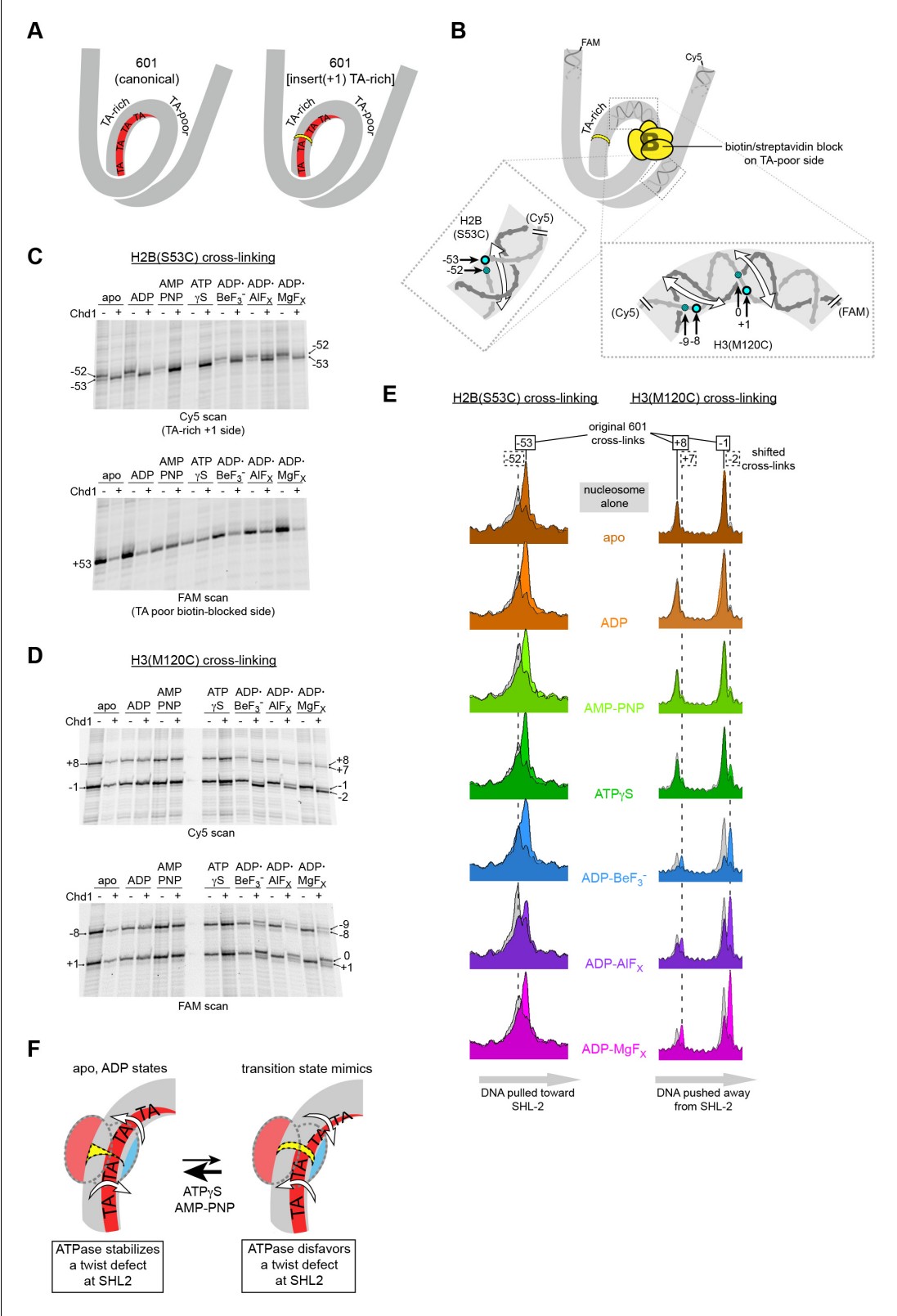

**Figure 6.** Chd1 binding induces a nucleotide-dependent twist in nucleosomal DNA at SHL2. (**A**) Schematic view of TA steps on the TA-rich side of the 601, which are phased in the canonical Widom 601 (left) and expected to be out-of-phase upon addition of a single bp at position 22 (yellow highlight, right). (**B**) Experimental design for monitoring histone-DNA contacts of the 601[insert(+1) TA-rich] nucleosomes. The presence of biotin/streptavidin at position 19 on the TA-poor side of the nucleosome will block Chd1 binding to SHL+2. Cross-linking of H2B(S53C) and H3(M120C) will report on the

*Figure 6 continued on next page*

*Figure 6 continued*

effects of Chd1 binding to SHL–2. In the expanded views, filled blue circles indicate the locations of cross-links as observed on the Cy5-labeled strand. For the H2B(S53C) view, the hatched blue circle denotes cross-links observed due to the +1 bp insertion on the TA-rich side as shown in (C), whereas for the H3(M120C) view, the hatched circles show the new dyad cross-links that appear with Chd1 bound in the presence of transition state analogs as shown in (D). Note that this design represents information gained from two different nucleosomes, each with a biotin/streptavidin block but with unique cross-linking sites. (C) The presence of Chd1 shifts entry DNA of 601[insert(+1) TA-rich] nucleosomes toward SHL–2 in all nucleotide conditions. Histone mapping reactions for biotinylated H2(S53C) nucleosomes plus streptavidin were carried out in the presence and absence of Chd1. The numbers indicate the cross-linking positions relative to the dyad, as observed for the canonical Widom 601 sequence. (D) Chd1 binding changes the position of DNA around the dyad of 601[insert(+1) TA-rich] nucleosomes in a nucleotide-dependent fashion. Cross-linking reactions were performed with biotinylated H3(M120C) nucleosomes in the presence of streptavidin. Gels shown in (C) and (D) are representative of four or more experiments. (E) Chd1 alters the distance between histone-DNA cross-links in a nucleotide dependent manner. Shown are intensity plots of each gel lane shown for Cy5 scans in (C) and (D). Nucleosome alone lanes are shown in gray, overlaid with the positions of cross-links observed for each nucleotide-bound state of Chd1 as colored peaks. The dotted lines indicate the positions of cross-links that are shifted 1 bp from those observed for the canonical 601, and correspond to the hatched blue circles shown in (B). The shift in cross-linking of H2B(S53C) from $-52$ to $-53$ corresponds with movement of entry DNA toward SHL–2, whereas the shift in H3(M120C) cross-links from $+8/-1$ to $+7/-2$ corresponds with movement of dyad DNA away from SHL–2. (F) A cartoon interpretation of Chd1-dependent changes in nucleosomal DNA.

DOI: https://doi.org/10.7554/eLife.34100.022

The following figure supplements are available for figure 6:

**Figure supplement 1.** Chd1 binding shifts DNA at the dyad of 601[insert(+1) TA-rich] in a nucleotide-dependent fashion.

DOI: https://doi.org/10.7554/eLife.34100.023

**Figure supplement 2.** Extended gel images of H2B and H3 mapping reactions of 601[insert(+1) TA-rich] nucleosomes containing a biotin/streptavidin block.

DOI: https://doi.org/10.7554/eLife.34100.024

H2B(S53C) cross-link by one nt, corresponding to DNA movement toward the remodeler. This shift resulted in a cross-link at the original $-53$ site of the 601 in spite of the 1 bp insertion (*Figure 6C,E*). In contrast, distinct nucleotide-dependent patterns were observed for H3(M120C), which cross-linked on the other side of SHL–2 (*Figure 6D*). In apo, ADP, and AMP-PNP conditions, the dyad cross-links of 601[insert(+1) TA-rich] were unchanged by Chd1, whereas with ADP·BeF$_3^-$, Chd1 stimulated a shift of the dyad cross-links by 1 bp. This shift corresponded to movement of DNA away from the the +1 insertion site on the TA-rich side. To further examine how ATP-mimics influence Chd1, we also tested ATPγS and the transition state analogs ADP·AlF$_X$ and ADP·MgF$_X$. Both transition state analogs behaved similarly to ADP·BeF$_3^-$, with cross-link shifts corresponding to DNA being pushed away from the SHL–2 site where the remodeler bound (*Figure 6D*; *Figure 6E*).

We interpret these cross-linking patterns as Chd1 promoting two types of DNA structures that either stabilize or disfavor a small DNA bulge at SHL–2 (*Figure 6F*). In apo and ADP conditions, Chd1 shifted the H2B(S53C) cross-link by one nt but had no effect on dyad cross-links, consistent with creating a small bulge that effectively absorbed the +1 bp insertion at SHL–2. With ADP·BeF$_3^-$ and the transition state analogs, Chd1 shifted cross-links at both H2B(S53C) and H3(M120C), thereby maintaining separation of these sites expected for the +1 bp insertion. We interpret the effect of Chd1 with ADP·BeF$_3^-$ and transition state analogs as stabilizing the canonical structure of nucleosomal DNA, in agreement with cryoEM structures of Chd1-nucleosome complexes (*Farnung et al., 2017*; *Sundaramoorthy et al., 2018*).

In addition to the predominant 1 bp shift at the dyad, significant levels of cross-linking in ADP·BeF$_3^-$, ADP·AlF$_X$ and ADP·MgF$_X$ were also observed at the original 601 position. The shifted and non-shifted cross-links may reflect an equilibrium of the bulged and non-bulged states of DNA at SHL2. Interestingly, ATPγS gave a small but reproducible cross-link representing the +1 shift at the dyad. While one interpretation is that the ATP-bound Chd1 may support either bulged or non-bulged DNA at SHL2 (*Figure 6F*), the dyad shift with ATPγS was more clearly apparent on one strand than the other, perhaps reflecting an intermediate structure.

## Discussion

SF1 and SF2 translocases travel along nucleic acids using an inchworm-type mechanism: opening of the bi-lobed ATPase motor in the apo state extends contacts by one nt in the direction of translocation, and domain closure upon ATP binding ratchets the motor along the nucleic acid substrate by

one nt (reviewed in *Singleton et al., 2007*). Here we describe a related but distinct process for DNA translocation on the nucleosome, where DNA movement occurs during both open and closed states of the motor.

In the inchworm model, the front and back halves of the motor take turns shifting past the nucleic acid substrate. For translocases like chromatin remodelers that move in a 3' to 5' direction, lobe 2 is the leading half, reaching forward in the open state to engage with DNA ahead of the motor. As monitored by site-specific cross-linking, lobe 2 of the Chd1 remodeler changed contacts with DNA in a nucleotide-dependent fashion and was highly sensitive to DNA sequence (*Figure 1*). Remarkably, the open state of the Chd1 ATPase showed the capability of pulling DNA toward itself. On the TA-poor side of the Widom 601 nucleosome, Chd1 shifted a large segment of DNA across the histone surface by 1–3 nt. This DNA shift toward the remodeler occurred in apo and ADP-bound states and was independent of nucleotide hydrolysis (*Figures 2* and *3*). We propose that such a shift could be accomplished if the open conformation of the ATPase motor trapped a small ~1 bp bulge of DNA. A bulge or DNA distortion at SHL2 could induce a concerted, corkscrew-like shift of DNA toward the remodeler. Recently, the cryoEM structure of the SWI/SNF ATPase bound to the nucleosome provided a first view of how an open ATPase motor engages with the nucleosome (*Liu et al., 2017*). This structure, which preferentially bound the TA-poor SHL+2 site of the Widom 601 nucleosome, revealed significant distortions in DNA at the binding site, with 3–6 Å displacements of the sugar-phosphate backbone from its canonical path and some disruptions in base stacking interactions. Such disruptive interactions are consistent with an ability to favor a ~ 1 bp bulge at SHL2 when both lobes of the ATPase motor bind in an open, active state.

The nucleotide-free states of SF1 (RecBCD) and SF2 (NS3) ATPases were previously found to unwind DNA duplexes, independently of nucleotide-driven translocation (*Farah and Smith, 1997*; *Levin et al., 2005*; *Singleton et al., 2004*). For NS3, it was proposed that a nucleotide-free unwinding step is achieved by favorable protein-DNA interactions, with the motor acting as a Brownian ratchet to capture DNA in a transiently melted state (*Levin et al., 2005*). For Chd1, instead of duplex melting, the open state of the ATPase motor appears to promote a bulged DNA duplex.

Compared to the open apo and ADP-bound states of the motor, nucleotide-bound states of Chd1 containing ATP mimics and transition state analogs favored distinct organizations of nucleosomal DNA at SHL2. With $ADP \cdot BeF_3^-$ and transition state analogs, Chd1 appeared to reinforce the canonical structure of nucleosomal DNA at SHL2, without a bulge. As shown with the 601[insert(+1) TA-rich] nucleosome variant, Chd1 binding at SHL–2 shifted DNA over the dyad by ~1 bp with transition state analogs, indicating an incompatibility with the SHL–2 bulge favored by apo and ADP-bound states (*Figure 6*). These observations agree with Chd1-nucleosome complexes bound to $ADP \cdot BeF_3^-$ and visualized by cryoEM, which show only modest perturbations of nucleosomal DNA at the bound SHL2 sites (*Farnung et al., 2017*; *Sundaramoorthy et al., 2018*).

Like ISWI- and SWI/SNF-type remodelers, Chd1 binds at SHL2 and translocates DNA toward the nucleosome dyad (*Farnung et al., 2017*; *McKnight et al., 2011*; *Nodelman et al., 2017*; *Sundaramoorthy et al., 2018*). We propose that the core mechanism for translocating nucleosomal DNA past the histone core arises from the ATPase motor enforcing unique geometries of DNA at its binding site (*Figure 7*). According to this model, DNA translocation is initiated by the open form of the remodeler ATPase motor binding at SHL2, which draws ~1 bp into a bulge and results in DNA on the entry gyre of the nucleosome being shifted toward the remodeler by ~1 bp. Subsequent closure of the ATPase cleft upon ATP binding and hydrolysis forces a redistribution of the bulged DNA, with collapse of the bulge pushing ~1 bp onto the DNA segment toward the dyad, on the other side of the motor. In this manner, the segments of DNA on either side of SHL2 alternately shift toward and away from the motor, with transitions between open and closed forms of the ATPase effectively transferring ~1 bp from one DNA segment to the other in the process.

While our data reveal that the ATPase motor can stabilize two distinct conformations of nucleosomal DNA that differ by ±1 bp, the ATPase motor may also favor intermediate DNA conformations, particularly in an ATP-bound state. The ATP analog states of Chd1 showed characteristics of apo/ADP and also $ADP \cdot BeF_3^-$, depending on the context. For lobe 2 cross-linking, AMP-PNP conditions were more similar to apo and ADP and different from $ADP \cdot BeF_3^-$ (*Figure 1*), yet for experiments showing movement of DNA on the TA-poor side of the 601, no effect was seen for Chd1 with AMP-PNP nor $ADP \cdot BeF_3^-$, unlike apo/ADP states (*Figure 2*). In the context of the 601[insert(+1) TA-rich] nucleosome, however, the Chd1-stimulated shifts of nucleosomal DNA with AMP-PNP and ATPγS

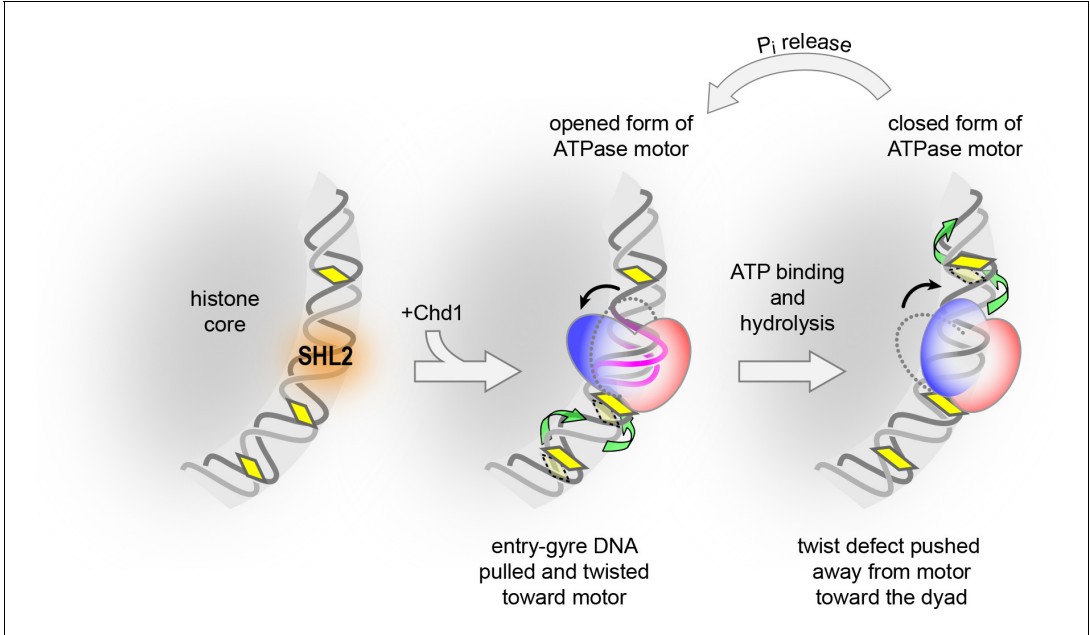

**Figure 7.** A model for DNA translocation on the nucleosome. Binding of the ATPase motor in an open state stabilizes a twist defect at SHL2, which pulls entry-side DNA toward the remodeler in a corkscrew-like motion. The closed state of the motor attained during ATP hydrolysis is incompatible with the twist defect at SHL2, forcing the extra DNA toward the dyad. Cycling of the ATPase motor therefore both creates twist defects and stimulates twist diffusion around the histone core.

DOI: https://doi.org/10.7554/eLife.34100.025

predominantly resembled apo and ADP, though a minor cross-link with ATPγS hints that nucleosomal DNA may occupy an intermediate state between bulged and non-bulged structures (*Figure 6*). The ATP-bound state may accommodate a range of local DNA perturbations and thus bridge the transition from bulged to non-bulged conformations of nucleosomal DNA. Future high resolution studies will be necessary for defining how intermediate states of the ATPase motor alter and respond to the structure and energetics of nucleosomal DNA during translocation.

Nucleosomes are expected to respond to DNA translocation by twist diffusion. Our data reveal that twist diffusion also defines the core mechanism by which remodelers shift nucleosomal DNA. For DNA to be ratcheted forward, nucleotide-dependent conformations of the ATPase motor must be coupled to pre- and post-shifted positions of nucleosomal DNA. Our data indicate that the pre- and post-shifted states correspond to distinct twist states of nucleosomal DNA. By accommodating ~1 additional bp, the DNA bulge caused by the apo/ADP states of the ATPase motor results in DNA undertwisting at the binding site (using the definition that more bp/turn means less twist per bp). Returning DNA conformation to a more canonical form, favored by restructuring of the motor as it hydrolyzes ATP, forces the twist defect onto the neighboring DNA segment. In this manner, distinct twist states on the nucleosome oscillate in response to ATP-driven changes of the motor, resulting in DNA being effectively pumped toward the dyad. This model differs from the conventional expectations of inchworm-type translocation, where DNA on either side of the motor coordinately moves as a unit during translocation. Instead, with twist defects the DNA moves as two alternating segments. As a Brownian ratchet, the motor achieves directionality through the ATP hydrolysis cycle yet relies on thermal fluctuations to transport DNA. By generating a twist defect, the motor uncouples the two segments of DNA, allowing the segment in front of the motor to shift without requiring simultaneous movement of the DNA segment on the exiting side of the motor. Subsequent elimination of the twist defect upon ATP hydrolysis presumably locks in the shifted position of entry DNA, necessary for directional diffusion of DNA away from the motor.

One consequence of action via twist defects is that the motor need not have a strong external contact for shifting DNA around the histone core. In the traditional inchworm model, a translocase would need to be held in place so that its attempt to walk along nucleosomal DNA would result in

pulling DNA in a corkscrew motion relative to the histone core. By creating twist defects, the DNA in front of the remodeler would shift relative to the DNA behind, with the remodeler maintaining contact with both portions of DNA during alternating movement. Although remodelers contact the H4 tail and interact with histone H3 via Snf2-specific insertions when bound at SHL2 (*Farnung et al., 2017*; *Liu et al., 2017*; *Sundaramoorthy et al., 2018*), we predict that these contact are likely not necessary for force generation and instead are important for properly orienting and positioning the motor at SHL2.

Nucleosomes have long been recognized to satisfy the core requirement of twist diffusion. The first and many subsequent crystal structures of the nucleosome showed that ±1 bp differences in DNA length readily occur at SHL2 and SHL5 (*Luger et al., 1997*; reviewed in *McGinty and Tan, 2015*; and *Muthurajan et al., 2003*). Biochemically, it was shown that these differences in DNA length, corresponding to twist defects, are dynamic and easily diffuse to other SHLs on the nucleo-some (*Edayathumangalam et al., 2005*). Twist diffusion has recently been visualized in molecular simulations of nucleosomes, where DNA segments bounded by minor groove contacts spontane-ously gain or lose single bps in coordination with adjacent DNA (*Brandani et al., 2018*). In addition to changes in DNA, cryoEM structures have recently shown that deformations of the histone core can also be coupled to distortions at SHL2 (*Bilokapic et al., 2018b*). In these structures, histone dimers appeared to pivot to maintain contact with displaced DNA, and in some cases formed dis-tinct histone-DNA contacts at SHL0.5 and SHL1.5. The ability of SHL2 to easily accommodate ±1 bp may be why Chd1 and other chromatin remodelers utilize this site for shifting DNA around the his-tone core. While defects can form spontaneously at SHL2, the nucleotide-dependent conformations of the ATPase motor both accelerate and add directionality to this process, first creating/capturing individual DNA bulges and then expelling excess DNA toward the dyad.

As Brownian ratchets, we anticipate that the actions or efficiency of remodelers may be impacted by the ease that twist defects can form at SHL2. We found a correlation between Chd1-dependent formation of a bulge at SHL2 and the sequence of the DNA segment that shifts toward the motor. Previously, Widom proposed that DNA sequence would affect DNA diffusion around the histone core, with faster DNA movement predicted for sequences that twist more easily (*Widom, 2001*). Our results are in agreement with this idea, and indicate that the ease of forming a bulge at SHL2 is coupled to the strength of phasing for the SHL2.5/SHL3.5 segment. With strong phasing, as seen with the periodic TA-steps, a corkscrew-like displacement of DNA is more expensive and disfavors the SHL2 bulge (*Figure 4*). We speculate that the influence of DNA sequence on bulge formation could impact remodeling activity. We previously showed that Chd1 preferentially shifts 601 nucleo-somes toward the TA-poor side, and that swapping the SHL2.5/SHL3.5 segment (24 to 39 bp) on each side of the 601 resulted in a more even distribution toward TA-rich and TA-poor sides (*Winger and Bowman, 2017*). Interestingly, nucleosome sliding was also perturbed when the 24–39 bp region was replaced by poly[dA:dT]. Although the full impact of this poly[dA:dT] segment on nucleosome structure and dynamics is not presently clear, this substitution produced several phased positions of neighboring DNA segments differing by ~1 bp, which could potentially influence forma-tion or directional elimination of a twist defect at SHL2 (*Winger and Bowman, 2017*). Future studies with other 601 variants and distinct positioning sequences will be necessary to clarify how particular DNA motifs on the nucleosome impact DNA translocation by chromatin remodelers. By proposing that DNA translocation by Chd1 is intimately tied to twist diffusion, the work described here should stimulate further investigations of both sequence-dependent repositioning and the basic mechanism of nucleosome sliding.

# Materials and methods

## Protein constructs, expression, and purification

All Chd1 constructs spanned residues 118–1274 of *Saccharomyces cerevisiae* Chd1. The 118–1274 portion includes the chromodomains, ATPase, and DNA-binding domains, as previously described (*Patel et al., 2011*) and is referred to throughout this manuscript as Chd1. A cysteine-free version of Chd1 was used as a template to introduce single cysteine mutations at specific positions: N459C, E493C, N650C, V721C, and S770C (*Nodelman et al., 2017*). The W793A substitution was intro-duced through site directed mutagenesis into the cysteine variant Chd1(N459C), in which all native

cysteines were mutated to alanine, and separately into an otherwise wildtype Chd1 background. All constructs were His-tagged, and were recombinantly expressed in *E. coli* and purified as described previously (*Patel et al., 2011*). Briefly, cells were lysed via sonication, clarified, and the cell extract loaded onto a HisTrap HP (GE Healthcare) column. After washing, His-tagged Chd1 was eluted using 175 mM imidazole and then further separated from contaminants via a HiTrap SP-FF cation exchange column (GE Healthcare). Following removal of the His-tag by overnight digestion with Precission protease, Chd1 was cleaned up by a second passage over a HisTrap HP column (collected in the flow through) and further purified using a HiLoad 16/600 Superdex 200 column (GE Healthcare).

The expression constructs for H3(C110A) and wild type H2A, H2B, and H4 *Xenopus laevis* histones were kindly provided by Karolin Luger. Single cysteine histone variants were generated by site-directed mutagenesis: H2A(G28C), H2A(A45C), H2B(R30C), H2B(S53C), H2B(T85C), H2B(T87C), and H3(M120C). H3(M120C) was made in the H3(C110A) background, which removed the native cysteine. Previous work by the Bartholomew group reported histone cross-linking for three sites used here: H2B(S53C) and H2A(A45C) (*Kassabov et al., 2002*), and H3(M120C) (*Hota et al., 2013*). All histones were expressed and purified as previously described (*Luger et al., 1999*), and stored as 2 mg lyophilized aliquots at −20 ˚C.

## Nucleosome constructs, reconstitution and purification

Histones were combined into tetramers, dimers or octamers by refolding equimolar amounts of the appropriate histones and then purified using a HiLoad 16/600 Superdex 200 size exclusion column as previously described (*Luger et al., 1999*; *Nodelman et al., 2017*). DNA fragments were generated via PCR using two primers that contained different fluorophores, thus resulting in doubly-labeled DNA. The template used for PCR was either the canonical Widom 601 sequence (*Lowary and Widom, 1998*), a 601 variant sequence (*Figure 1—figure supplement 2*), or the 5S nucleosome positioning sequence (*Figure 5—figure supplement 1*). Nucleosome constructs used for sliding reactions were end-positioned, with the 145 bp Widom 601 sequence flanked by 80 bp on the TA-rich side and 0 bp on the TA-poor side (called 80N0). Biotinylated DNA was produced using long primers containing an internal dT-biotin located 19 nt and 5′ relative to the 601 dyad. On the TA-rich side, the biotinylated construct was FAM-19-(+biotin)601-29-Cy5, with biotin on the FAM strand. On the TA-poor side, two constructs were made: Cy5-40-601(+biotin)-19-FAM and Cy5-40-[insert(+1) TA-rich]601(+biotin)-19-FAM, each with biotin on the FAM strand. All other nucleosome constructs were Cy5-40N40-FAM. For each construct, the desired DNA product was purified away from primers by separation on 6% (60:1) acrylamide:bisacrylamide native gels.

To produce nucleosomes, each DNA fragment was first combined in a 1:1 ratio with octamer, or a 1:1:2.2 ratio with tetramer and dimer in 2M KCl. To deposit the histone octamer on the nucleosome positioning sequence, the DNA and histone mixture was gradually dialyzed from 2 M to 200 mM KCl at 4˚C over a 24 hr period, with a final transfer to 2.5 mM KCl buffer (*Luger et al., 1999*). Centered nucleosomes were purified away from free DNA, hexasomes, and off-positioned nucleosome species by separation on 7% (60:1) acrylamide:bisacrylamide native gels.

## Chd1 site-specific cross-linking

Chd1 constructs containing a single cysteine were diluted to a concentration of 7.5 µM and labeled in the dark at room temperature with 4-azidophenacyl bromide (APB, dissolved in DMF; Sigma, cat# A6057) for 2.5 hr, $C_f$ = 400 µM. Labeling and cross-linking to the nucleosome were previously described (*Nodelman et al., 2017*). Briefly, samples to be crosslinked were assembled in 1x Slide buffer (20 mM Tris-HCl, pH 7.5, 50 mM KCl, 5 mM MgCl$_2$, 0.1 mg/ml bovine serum albumin (BSA), 5 mM dithiothreitol (DTT), 5% sucrose) for either apo, 2 mM ADP, or 2 mM AMPPNP; or slide buffer minus magnesium for ADP·BeF$_3^-$ (2 mM ADP, 15 mM NaF, 3 mM BeCl$_2$, and 6 mM MgCl$_2$). Nucleotide was added to tubes, followed by either nucleosome or naked DNA (naked DNA was equivalent to that used in nucleosome reconstitutions) to a final concentration of 150 nM, and finally APB-labeled Chd1 ($C_f$ = 450 nM). Chd1 was incubated with nucleosome in the dark for 30–45 min, UV irradiated for 15 s (λ = 302 nm; VWR UV transilluminator 89131–440), followed by the addition of 100 µl quench (20 mM Tris-HCl, pH 7.5, 50 mM KCl, 0.1 mg/mL BSA, 5 mM EDTA, and 5 mM DTT) and 150 µl post-irradiation buffer (20 mM Tris-HCl, pH 8.0, 0.15% SDS, 50 mM NaCl). Samples were processed as described (*Kassabov and Bartholomew, 2004*; *Nodelman et al., 2017*).

## Histone mapping

Histone mapping followed the published methods of Kassabov and Bartholomew (*Kassabov and Bartholomew, 2004*). Nucleosomes were first buffer exchanged into cold 20 mM Tris-HCl, pH 7.5, 5% glycerol to remove DTT, and then labeled with ~220 μM APB in the dark for ~2.5 to 3 hr at room temperature. During nucleosome labeling, sample tubes were prepared at room temperature in the absence (apo) or presence of 2 mM nucleotide in 1X Slide buffer. ADP·BeF$_3^-$, ADP·AlF$_X$, and ADP·MgF$_X$ conditions all contained 2 mM ADP, 15 mM NaF, and 6 mM MgCl$_2$, with ADP·BeF$_3^-$ and ADP·AlF$_X$, also containing either 3 mM BeCl$_2$ or 3 mM AlCl$_3$, respectively. For experiments with 5S nucleosomes (*Figure 5*), ADP stocks (100 mM) were pre-treated to remove residual ATP by incubation with 100 Units/ml of *S. cerevisiae* hexokinase (Sigma-Aldrich, cat# H4502), 1 M glucose, and 5 mM MgCl$_2$ for 20 min at 25°C. Where indicated, Chd1 was added to a final concentration of 600 nM immediately before nucleosomes. After labeling, nucleosomes were added to the sample tubes to a final concentration of 150 nM in the dark. The samples were incubated for 10–30 min and then given a 15 s exposure to UV light to induce cross-linking. After UV exposure, samples were quenched 2:1 with quench buffer, diluted 1:1 with post-irradiation buffer (20 mM Tris-HCl, pH 8.0, 0.15% SDS, 50 mM NaCl), and then heated at 70°C for 20 min. Subsequent phenol clean up of cross-linked DNA and NaOH cleavage steps were followed as described (*Kassabov and Bartholomew, 2004*; *Nodelman et al., 2017*). DNA samples, resuspended in deionized formamide loading buffer, were separated on an 8 M urea denaturing gel (8% (19:1) acrylamide:bisacrylamide) and run at 65 Watts, typically 1.5 to 3 hr depending on fragment size. The DNA fragments were visualized via their fluorophores on a 9410 Typhoon variable mode imager (GE).

Experiments using biotinylated nucleosomes were APB-labeled on either H2B-S53C or H3-M120C. Nucleosomes for *Figure 3* were incubated at room temperature for 15 min with a final concentration of either 9.6 μM streptavidin (Pierce, cat# 21125), a 64-fold excess over nucleosome (150 nM), or 1x Slide buffer for the no streptavidin samples. For *Figure 6* experiments, all samples were pre-incubated with 9.6 μM streptavidin under the same conditions as *Figure 3* in the presence of nucleotide analog or apo conditions. Chd1 was then added where appropriate to a final concentration of 600 nM (4-fold over nucleosome), and 50 μl samples were further incubated for 20 min. Samples were UV irradiated and processed as described above for histone mapping.

## Native gel sliding assay

Nucleosome sliding assays were conducted similar to *Patel et al. (2011)*. Remodeling reactions (20 μl) were performed at room temperature with 50 nM of either Chd1 or Chd1(W793A) and 150 nM FAM-labeled 80-601-0 nucleosomes in 1x Slide buffer (20 mM Hepes-KOH, pH 7.6, 50 mM KCl, 5 mM MgCl$_2$, 0.1 mg/ml BSA, 1 mM DTT, 5% sucrose). After removal of a t = 0 time point, time-keeping was started upon addition of 2.5 mM ATP, with 1 μl of remodeling reaction transferred to 20 μl quench buffer (20 mM Hepes-KOH, pH 7.6, 50 mM KCl, 0.1 mg/ml BSA, 1 mM DTT, 5% sucrose, 5 mM EDTA, pH 8.0, and 150 ng/μl ultrapure salmon sperm DNA (Thermo Fisher Scientific, cat#15632011)) and tubes placed on ice. Time point samples were loaded onto 6% (60:1) acrylamide:bis-acrylamide native PAGE gels and run at 100V in 0.25xTBE for 100 min. Gels were scanned on a 9410 Typhoon variable mode imager (GE) and quantitated using ImageJ software (RRID:SCR_003070). Using *Mathematica* (RRID:SCR_014448), data were fit to the equation y = a$_1$*(1-exp(-k$_1$*x)) + a$_2$*(1-exp(-k$_2$*x))+c, where k$_1$ and k$_2$ are apparent rate constants, a$_1$ and a$_2$ are corresponding amplitudes, and c is a constant.

## Acknowledgements

We thank Michael McCaffery and the Integrated Imaging Center (IIC) at Johns Hopkins University for equipment use. This work was funded by NIH grant R01-GM084192 (to GDB).

## Additional information

### Funding

| Funder | Grant reference number | Author |
|---|---|---|
| National Institutes of Health | R01-GM084192 | Jessica Winger<br>Ilana M Nodelman<br>Robert F Levendosky<br>Gregory D Bowman |

The funders had no role in study design, data collection and interpretation, or the decision to submit the work for publication.

### Author contributions

Jessica Winger, Conceptualization, Investigation, Visualization, Writing—review and editing, Reagent purification, Discovery of Chd1-dependent shift in histone cross-linking, Data acquisition for Figures 2, 4, and 5; Ilana M Nodelman, Conceptualization, Investigation, Visualization, Writing—review and editing, Reagent purification, Data acquisition for Figures 1, 3, and 6; Robert F Levendosky, Conceptualization, Investigation, Writing—review and editing, Reagent purification; Gregory D Bowman, Conceptualization, Supervision, Funding acquisition, Visualization, Writing—original draft, Writing—review and editing

### Author ORCIDs

Jessica Winger http://orcid.org/0000-0003-2109-3560
Robert F Levendosky http://orcid.org/0000-0002-5101-0810
Gregory D Bowman http://orcid.org/0000-0001-8025-4315

### Decision letter and Author response

Decision letter https://doi.org/10.7554/eLife.34100.028
Author response https://doi.org/10.7554/eLife.34100.029

## Additional files

### Supplementary files

• Transparent reporting form
DOI: https://doi.org/10.7554/eLife.34100.026

### Data availability

Extended gel images are given in figure supplements.

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
