## [Decision Letter]

Thank you for submitting your article "A two-step process for translocating nucleosomal DNA" for consideration by *eLife*. Your article has been reviewed by three peer reviewers, and the evaluation has been overseen by a Reviewing Editor and Kevin Struhl as the Senior Editor. The reviewers have opted to remain anonymous.

The reviewers have discussed the reviews with one another and the Reviewing Editor has drafted this decision to help you prepare a revised submission.

Summary:

In this manuscript site directed cross linking is used to study how DNA is redistributed on nucleosomes bound by the remodeling enzyme Chd1. Consistent with previous studies the ATPase domains of Chd1 are found to bind nucleosomes in the region 2 turns away from the dyad. Changes in the interactions of Chd1 are observed to occur following addition of some combinations of nucleotides. The interaction of histones with DNA is also monitored following binding with Chd1 and different nucleotides. Surprisingly, different changes to histone DNA contacts are observed on each side of the nucleosome. The authors explore this further by generating a series of nucleosomes with different DNA sequences. This clearly shows that the structural and or sequence properties of DNA influence how Chd1 alters histone DNA interactions.

The key observations are that in the apo or ADP states, Chd1 can generate significant changes in the way DNA is placed across the octamer surface. These differ in the AMP-PMP or ADP·BeF_3_^-^ states consistent with the ability of Chd1 to generate distortions to DNA during the course of cycles of ATP hydrolysis. The study also highlights the fact that the way in which Chd1 and likely other remodeling enzymes affect nucleosomal DNA is affected by DNA sequence. These are important observations that further understanding of how remodeling ATPases function.

Essential revisions:

1) H2A(G28C) (+42) and H2A(A45C) (+/-40) cross-linking is detected (Figure 2—figure supplement 3) but no NTP dependent changes are observed. This is very difficult to reconcile with the changes observed via H2B(S53C) at +53. The authors suggest that this may arise from sequence specificity of the cross-linking reaction. This is an important point, there is likely a limit to the extent that cross-linking can be interpreted as local conformation may preclude an orientation favorable for cross-linking even when the distance is within range. As a result the technique may not give genuine single nucleotide precision. Through most of the manuscript this aspect is treated with due caution, for example interpreting changes as 1 or 2 bp. However, the lack of Chd1 dependent effects at H2A(G28C) and H2A(A45C) are ignored in favor of the changes observed at the one site where Chd1 dependent changes are observed (H2B(S53C)). This is a key point central to the manuscript so further study is required to address this.

Through use of steric occlusion by streptavidin it is elegantly shown that the nucleotide dependent changes at +53 result from Chd1 binding at SHL+2, and that these do not propagate across the dyad as cross-linking to M120C is not affected. However, in other experiments, especially those performed with the insertion at +1, different nucleotide dependent changes are observed predominantly affecting how DNA is arranged across the dyad (Figure 4). In this case these changes cannot be attributed to Chd1 binding at one side or the other and interpretation is as a result complex. It would be ideal to perform this experiment after streptavidin occlusion.

It needs to be shown that Chd1 binds to both sides of the +1 insert nucleosomes.

2) Is the proposed bulge consistent with the bulge observed by Liu et al., 2017? And is the nucleotide state in which the bulge can be observed the same in these two studies?

I wonder whether the change in the H2B cross-links in different nucleotide states and the bulge around SHL2 are due to the 601 sequence. To address this concern, can the authors repeat key experiments with a different DNA sequence? Maybe a chimeric DNA sequence that contains the TA-rich half of the 601 would be useful for this purpose.

3) Experiments are all interpreted through the (reasonable) idea that a bulge is stabilized. Is it also formally possible that Chd1 binding might deform the nucleosome through altered octamer conformation resulting in change to specific crosslink positions for H2B(S53C)? Deeper discussion of alternatives to the bulge need to appear in the main text. The discussion of this is currently in the supplementary discussion but should be brought into the main text and discussion of alternatives that don't involve changes in DNA wrapping.

4) The schematics in Figure 4 should be altered to make it more accessible. We found it difficult to follow the directionality with which DNA was repositioned from the schematics shown in Figure 4. Can this and the numbering system used for the DNA with the 1bp insert be explained more clearly? In Figure 4C, I'm not sure whether the two illustrations of the dyad and H2B cross-links are necessary. Isn't one just the reverse of the other? Redundancy often confuses the reader. I suggest simplifying this panel, defining one of these two states as the ground state.

---

## [Author Response]

Essential revisions:1) H2A(G28C) (+42) and H2A(A45C) (+/-40) cross-linking is detected (Figure 2—figure supplement 3) but no NTP dependent changes are observed. This is very difficult to reconcile with the changes observed via H2B(S53C) at +53. The authors suggest that this may arise from sequence specificity of the cross-linking reaction. This is an important point, there is likely a limit to the extent that cross-linking can be interpreted as local conformation may preclude an orientation favorable for cross-linking even when the distance is within range. As a result the technique may not give genuine single nucleotide precision. Through most of the manuscript this aspect is treated with due caution, for example interpreting changes as 1 or 2 bp. However, the lack of Chd1 dependent effects at H2A(G28C) and H2A(A45C) are ignored in favor of the changes observed at the one site where Chd1 dependent changes are observed (H2B(S53C)). This is a key point central to the manuscript so further study is required to address this.

In the original submission, we concluded that H2A(G28C) did not robustly respond to Chd1, since the signal we consistently saw was barely above background. We have been continually trying to improve our mapping reactions, and were fortunate to find that with higher sensitivity we can detect a robust Chd1-dependent signal from H2A(G28C). This cross-linker is therefore now included in Figure 2. We believe that this issue highlights a weakness of the cross-linking technique: given its extreme sensitivity, the absence of a change in cross-linking is a negative result and therefore should not be overinterpreted as no change in structure. With this in mind, we have tried to remain cautious in interpretations of cross-linking in the text. We clearly see strong effects of DNA sequence on cross-linking, and as we state in the text, differences in DNA sequence may explain an inability to detect changes in other cross-linking sites, such as H2B(T85C) and H2A(A45C). We also point out, though, that all three cross-linking sites we report – H2B(S53C), H2A(G28C), and H2B(T87C) – show Chd1-dependent changes on the same DNA strand, corresponding to the tracking strand. H2A(A45C) cross-links to the guide strand, and this raises the possibility that the differences in cross-linking could reflect strand-specific responses to Chd1 binding.

Through use of steric occlusion by streptavidin it is elegantly shown that the nucleotide dependent changes at +53 result from Chd1 binding at SHL+2, and that these do not propagate across the dyad as cross-linking to M120C is not affected. However, in other experiments, especially those performed with the insertion at +1, different nucleotide dependent changes are observed predominantly affecting how DNA is arranged across the dyad (Figure 4). In this case these changes cannot be attributed to Chd1 binding at one side or the other and interpretation is as a result complex. It would be ideal to perform this experiment after streptavidin occlusion.

To address these concerns, we generated new +1 insertion nucleosomes containing a biotin/streptavidin block to allow for greater clarity in interpreting results. Interestingly, there was a difference between the biotinylated and nonbiotinylated 601[insert (+1) TA-rich] nucleosomes, specifically for the AMP-PNP conditions, which suggests that the shifts at the dyad in AMP-PNP were likely due to Chd1 binding on both sides. The experiments with the new biotinylated +1 nucleosomes revealed differences between AMP-PNP and ADP·BeF_3_^-^, and we therefore performed experiments with additional conditions (ATPgammaS and transition state analogs) to better understand the nucleotide-specific effects. These new experiments (Figure 6) indicate a difference between ATP-bound and ATP-hydrolysis/transition state conformations, and support our previous conclusion that Chd1 can alter the twist of DNA in a nucleotide-directed manner. The experiments with the non-biotinylated nucleosomes are now presented in Figure 6—figure supplement 1.

It needs to be shown that Chd1 binds to both sides of the +1 insert nucleosomes.

The surprising effect observed with the +1 insertion nucleosomes was that the dyad cross-linking shifted depending on the nucleotide-bound state of Chd1. Our new experiments with the biotin/streptavidin block on the TA-poor side (described in the paragraph above) also show this interesting effect. Since the biotin/streptavidin block determines where Chd1 can bind to the +1 nucleosomes, we did not further pursue the question of whether Chd1 could bind to both sides without a biotin/streptavidin block.

2) Is the proposed bulge consistent with the bulge observed by Liu et al., 2017? And is the nucleotide state in which the bulge can be observed the same in these two studies?

Yes, the cryoEM structure reported by Liu et al. (SWI/SNF ATPase bound to a nucleosome) was also in a nucleotide-free state and on the TA-poor side of the nucleosome. As we state in the Discussion, the DNA at SHL2 was found to be distorted in that structure, and the amplitude of those changes are consistent with an ability to allow an additional bp in this region. We also highlight how the structure of DNA for the apo-SWI/SNF is much more distorted than Chd1 bound to SHL2 in an ADP·BeF_3_^-^-bound conformation (Farnung et al., 2017; Sundaramoorthy et al., 2018).

I wonder whether the change in the H2B cross-links in different nucleotide states and the bulge around SHL2 are due to the 601 sequence. To address this concern, can the authors repeat key experiments with a different DNA sequence? Maybe a chimeric DNA sequence that contains the TA-rich half of the 601 would be useful for this purpose.

To address the question as to whether this Chd1-stimulated shift in DNA was unique to the 601 sequence, we also performed similar cross-linking experiments with the 5S positioning sequence (Figure 5). Similar to the TA-poor side of the Widom 601, experiments with 5S nucleosomes revealed changes in cross-linking in a Chd1- and nucleotide-dependent fashion. We obtained several distinct positions of 5S nucleosomes, and interestingly, some positions showed no changes. These results support that the subtle changes in DNA that we captured are sensitive to sequence, as we describe on the TA-poor and TA-rich sides of the 601 sequence.

3) Experiments are all interpreted through the (reasonable) idea that a bulge is stabilized. Is it also formally possible that Chd1 binding might deform the nucleosome through altered octamer conformation resulting in change to specific crosslink positions for H2B(S53C)? Deeper discussion of alternatives to the bulge need to appear in the main text. The discussion of this is currently in the supplementary discussion but should be brought into the main text and discussion of alternatives that don't involve changes in DNA wrapping.

We believe that the +1 insertion nucleosome strongly argues that Chd1 can change DNA twist at SHL2. In that experiment, there are four cross-links at the dyad (two on each strand), and all four show a predominant shift by 1 bp in ADP·BeF_3_^-^ and transition state analogs, but not other nucleotide states. A relative twist at SHL2 is consistent with the idea of a ~1 bp bulge, which can also explain the H2B(S53C), H2A(G28C) and H2B(T87C) shifts in cross-linking in apo/ADP states. We now explicitly state that the changes in entry DNA cross-linking could be due to movement of the H2A/H2B dimer or movement of DNA. However, combined with the +1 insert nucleosome experiments, we believe that the simplest explanation is DNA movement due to a DNA bulge at the ATPase binding site, which is known to accommodate +/- bp on the nucleosome. The potential for histone octamer distortion during remodeling adds an interesting dimension, but the vast majority of contacts from Chd1 are with nucleosomal DNA, not histones, and we believe that changes in the histone core would be a response to DNA distortion.

4) The schematics in Figure 4 should be altered to make it more accessible. We found it difficult to follow the directionality with which DNA was repositioned from the schematics shown in Figure 4. Can this and the numbering system used for the DNA with the 1bp insert be explained more clearly? In Figure 4C, I'm not sure whether the two illustrations of the dyad and H2B cross-links are necessary. Isn't one just the reverse of the other? Redundancy often confuses the reader. I suggest simplifying this panel, defining one of these two states as the ground state.

We appreciate the reviewers’ comments regarding the difficulty in interpreting these data. We have reworked the schematics for this figure (new Figure 6), which we believe should now more clearly show the how the nucleotide-dependent changes in DNA cross-linking positions relate to changes in DNA position and twist.